# An expanding manifold in transmodal regions characterizes adolescent reconfiguration of structural connectome organization

Bo-yong Park[1,2]*, Richard AI Bethlehem[3,4], Casey Paquola[1,5], Sara Larivière[1], Raul Rodríguez-Cruces[1], Reinder Vos de Wael[1], Neuroscience in Psychiatry Network (NSPN) Consortium[†], Edward T Bullmore[4], Boris C Bernhardt[1]*

[1]McConnell Brain Imaging Centre, Montreal Neurological Institute and Hospital, McGill University, Montreal, Canada; [2]Department of Data Science, Inha University, Incheon, Republic of Korea; [3]Autism Research Centre, Department of Psychiatry, University of Cambridge, Cambridge, United Kingdom; [4]Brain Mapping Unit, Department of Psychiatry, University of Cambridge, Cambridge, United Kingdom; [5]Institute of Neuroscience and Medicine (INM-1), Forschungszentrum Jülich, Jülich, Germany

*For correspondence:
by9433@gmail.com (B-P);
boris.bernhardt@mcgill.ca (BCB)

[†]A complete list of investigators from the Neuroscience in Psychiatry Network (NSPN) Consortium can be found in the Supplementary Information.

Group author details:
Neuroscience in Psychiatry Network (NSPN) Consortium See page 21

**Abstract** Adolescence is a critical time for the continued maturation of brain networks. Here, we assessed structural connectome development in a large longitudinal sample ranging from childhood to young adulthood. By projecting high-dimensional connectomes into compact manifold spaces, we identified a marked expansion of structural connectomes, with strongest effects in transmodal regions during adolescence. Findings reflected increased within-module connectivity together with increased segregation, indicating increasing differentiation of higher-order association networks from the rest of the brain. Projection of subcortico-cortical connectivity patterns into these manifolds showed parallel alterations in pathways centered on the caudate and thalamus. Connectome findings were contextualized via spatial transcriptome association analysis, highlighting genes enriched in cortex, thalamus, and striatum. Statistical learning of cortical and subcortical manifold features at baseline and their maturational change predicted measures of intelligence at follow-up. Our findings demonstrate that connectome manifold learning can bridge the conceptual and empirical gaps between macroscale network reconfigurations, microscale processes, and cognitive outcomes in adolescent development.

## Introduction

Adolescence is a time of profound and genetically mediated changes in whole-brain network organization (*Larsen and Luna, 2018*; *Menon, 2013*). Adolescent development is important for the maturation in cognitive and educational functions and brain health more generally, a notion reinforced by the overlapping onset of several neurodevelopmental and psychiatric disorders (*Hong et al., 2019*; *Khundrakpam et al., 2017*; *Paus et al., 2008*). With increased capacity to carry out longitudinal studies in large samples, it is now possible to track changes in brain network organization within subjects, providing insights into maturational processes, their biological underpinnings, and their effects on behavior and cognition.

By offering an in vivo window into brain organization, neuroimaging techniques, such as magnetic resonance imaging (MRI), offer the ability to track adolescent brain development over time. Several

 

cross-sectional and longitudinal studies in neurodevelopmental cohorts have focused on the analysis of morphological changes (*Gogtay et al., 2004*; *Shaw et al., 2006*; *Tamnes et al., 2017*), including MRI-based cortical thickness (*Shaw et al., 2006*; *Tamnes et al., 2017*) and volumetric measures (*Gogtay et al., 2004*; *Tamnes et al., 2017*). Studies robustly show initial gray matter increases until mid-late childhood followed by a decline for the rest of the lifespan. During adolescence, cortical thickness decreases in widespread brain regions (*Khundrakpam et al., 2013*; *Shaw et al., 2006*; *Sotiras et al., 2017*; *Tamnes et al., 2017*). Thus, contextualizing connectome alterations relative to established patterns of cortical thickness findings may establish whether inter-regional network changes occur above and beyond these diffuse effects of regional morphological maturation. More recent work explored changes in intracortical microstructure, capitalizing on myelin-sensitive contrasts such as magnetization transfer ratio (MT) mapping, which generally suggest overall increases in adolescence (*Paquola et al., 2019a*; *Whitaker et al., 2016*) together with depth-dependent shifts in intracortical myelin profiles (*Paquola et al., 2019a*). Besides the increasingly recognized changes in cortico-cortical connectivity, studying subcortical regions offer additional insights for understanding brain maturation during adolescence. Indeed, an increasing body of connectome-level studies emphasizes that subcortical structures contribute significantly to patterns of whole-brain organization, dynamics, and cognition (*Hwang et al., 2017*; *Müller et al., 2020*; *Shine et al., 2019*). In prior neurodevelopmental studies, it has been shown that the volumes of the striatum and thalamus decrease between adolescence and adulthood, potentially paralleling processes resulting in cortical gray matter reduction during this time window (*Herting et al., 2018*). A close inter-relationship between cortical and subcortical development is also suggested by recent functional connectivity work suggesting that cortico-subcortical pathways are intrinsically remodeled during adolescence (*Váša et al., 2020*), and these changes affect cognitive functioning. Collectively, these prior findings suggest measurable trajectories of cortical and subcortical structural organization and support associations to cognitive development (*Baum et al., 2020*; *Shaw et al., 2006*).

Recent conceptual and methodological advances enable the study of brain organization, development, and substrates underlying cognitive trajectories in humans. One key modality to track developmental changes in structural connectivity is diffusion MRI (dMRI), a technique sensitive to the displacement of water in tissue that allows for the non-invasive approximation of inter-regional white matter tracts. Prior cross-sectional and longitudinal studies in children and adolescents outlined changes in the microstructure of major white matter tracts during development based on the analysis of dMRI-derived tissue parameters (*Lebel and Beaulieu, 2011*; *Schmithorst and Yuan, 2010*). These findings have been complemented by assessments of brain network topology using graph-theoretical analysis (*Baker et al., 2015*; *Hagmann et al., 2010*; *Lebel and Beaulieu, 2011*; *Oldham and Fornito, 2019*), which reported a relatively preserved spatial layout of structural hubs across adolescent development on the one hand (*Hagmann et al., 2010*), yet with a continued strengthening of their connectivity profiles, likely underpinned by the ongoing maturation of long-range association fibers (*Baker et al., 2015*; *Lebel and Beaulieu, 2011*; *Oldham and Fornito, 2019*).

One emerging approach to address connectome organization and development comes from the application of manifold learning techniques to connectivity datasets. By decomposing whole-brain structural and functional connectomes into a series of lower dimensional axes capturing spatial gradients of connectivity variations, these techniques provide a compact perspective on large-scale connectome organization (*Margulies et al., 2016*; *Paquola et al., 2019b*; *Vos de Wael et al., 2020a*). In addition, these techniques capture multiple, potentially overlapping gradients in connectivity along cortical mantle, which can represent both subregional heterogeneity and multiplicity within a brain region (*Haak and Beckmann, 2020*). In prior work, we showed that multiple dMRI gradients can illustrate structural underpinnings of dynamic functional communication in the adult human connectome (*Park et al., 2021b*). In line with prior conceptual accounts, the low-dimensional eigenvectors (i.e., gradients) derived from these techniques provide continuous dimensions of cortical organization, and thus the eigenvectors can jointly generate intrinsic coordinate systems of the brain based on connectivity (*Bijsterbosch et al., 2020*; *Haak et al., 2018*; *Huntenburg et al., 2018*; *Margulies et al., 2016*; *Mars et al., 2018*). Beyond these methodological considerations, prior work has shown that the principal gradients estimated from resting-state functional (*Margulies et al., 2016*), microstructural (*Paquola et al., 2019b*), and diffusion MRI (*Park et al., 2021a*) all converge broadly along an established model of sensory-fugal hierarchy and laminar differentiation

(*Mesulam, 1998*), allowing gradient mapping techniques to make conceptual contact to theories of cortical organization, development, and evolution (*Buckner and Krienen, 2013*; *Goulas et al., 2018*; *Huntenburg et al., 2018*; *Sanides, 1969*; *Sanides, 1962*). An emerging literature has indeed shown utility of the gradient framework to study primate evolution and cross-species alignment (*Blazquez Freches et al., 2020*; *Valk et al., 2020*; *Xu et al., 2020*), neurodevelopment (*Hong et al., 2019*; *Paquola et al., 2019a*), as well as plasticity and structure-function coupling (*Park et al., 2021b*; *Valk Sofie et al., 2020*; *Vázquez-Rodríguez et al., 2019*). In a recent assessment by our team, manifold learning techniques have been applied to myelin sensitive intracortical MT data, showing an increasing myeloarchitectural differentiation of association cortex throughout adolescence (*Paquola et al., 2019a*). Still, the longitudinal maturation of dMRI connectomes in children and adolescents using manifold techniques has not been tracked.

Imaging-transcriptomics approaches allow for the identification of cellular and molecular factors that co-vary with imaging-based findings (*Arnatkeviciute et al., 2019*; *Fornito et al., 2019*; *Gorgolewski et al., 2014*; *Hawrylycz et al., 2015*; *Thompson et al., 2013*). Recently established resources, such as the Allen Human Brain Atlas (*Arnatkeviciute et al., 2019*; *Hawrylycz et al., 2015*), can be utilized to spatially associate macroscale imaging/connectome data with the expression patterns of thousands of genes. These findings have already been applied in the study of healthy adults (*Hawrylycz et al., 2015*; *Park et al., 2020*) and typically developing adolescents (*Mascarell Maričić et al., 2020*; *Padmanabhan and Luna, 2014*; *Paquola et al., 2019a*; *Vértes et al., 2016*; *Whitaker et al., 2016*), as well as individuals suffering from prevalent brain disorders (*Altmann et al., 2018*; *Hashimoto et al., 2015*; *Klein et al., 2017*; *Park et al., 2021a*; *Patel et al., 2021*; *Romero-Garcia et al., 2019*). The gene sets that co-vary with in vivo findings can furthermore be subjected to gene set enrichment analyses to discover potentially implicated molecular, cellular, and pathological processes (*Ashburner et al., 2000*; *Carbon et al., 2019*; *Chen et al., 2013*; *Dougherty et al., 2010*; *Kuleshov et al., 2016*; *Morgan et al., 2019*; *Romero-Garcia et al., 2018*; *Subramanian et al., 2005*). For example, studies in newborns have shown that cortical morphology reflects spatiotemporal patterns of gene expression in fetuses, linking molecular mechanisms to in vivo measures of cortical development in early life (*Ball et al., 2020*). Work in adolescents has furthermore shown that developmental changes in regional cortical thickness measures and myelin proxies spatially co-localize with the expression patterns of genes involved in synaptic and oligodendroglial function (*Paquola et al., 2019a*; *Whitaker et al., 2016*). Building on these prior investigations, the current study aimed at exploring whether adolescent structural connectome reconfigurations, assessed using manifold learning techniques, reflect the expression patterns of specific genes in order to identify potential molecular signatures of macroscale structural network development.

Here, we charted developmental changes in structural connectome organization, based on an accelerated longitudinal neuroimaging study involving 208 participants investigated between 14 and 26 years of age (*Kiddle et al., 2018*; *Whitaker et al., 2016*). Compared to cross-sectional designs, longitudinal studies track within-subject change, separating developmental effects from between-subject variability (*Louis et al., 1986*). We first estimated longitudinal changes in structural connectome manifolds across age. This compact and lower dimensional space furthermore allowed for the integration of connectome-level findings with changes in MRI-based measures of cortical morphology and intracortical myelin. We furthermore projected subcortico-cortical connectivity patterns into the manifold space to assess parallel developmental shifts of these pathways in the studied time window. Connectome manifold changes were contextualized at the molecular level via transcriptomic association and developmental enrichment analyses based on *post-mortem* datasets, which furthermore allowed for data-driven exploration of time windows of spatially co-localized gene sets. To also assess behavioral associations of connectome manifold changes, we utilized supervised machine learning to predict future measures of cognitive function quantified via the intelligence quotient (IQ). IQ is a widely used marker of general cognitive abilities, which shows good test–retest reliability (*Brown and May, 1979*; *Watkins and Smith, 2013*; *Catron, 1978*; *G.-Matarazzo et al., 1973*; *Snow et al., 1989*; *Wagner and Caldwell, 1979*) and has previously been applied to index overall cognitive function during development (*Crespi, 2016*; *Garde et al., 2005*; *Garde et al., 2000*; *Koenis et al., 2018*; *Park et al., 2016*; *Ramsden et al., 2011*; *Shaw et al., 2006*; *Suprano et al., 2020*). In the study of neurodevelopment, neuroimaging reports have previously assessed associations between IQ and large-scale network measures in children to adolescents (*Koenis et al., 2018*;

*Ramsden et al., 2011*; *NSPN Consortium et al., 2018*; *Shaw et al., 2006*; *Suprano et al., 2020*). Multiple sensitivity analyses were conducted at several steps to verify the robustness of our findings, and analytical code is made fully accessible to allow for independent replication of our findings.

## Results

These findings were based on the Neuroscience in Psychiatry Network (NSPN) cohort (*Kiddle et al., 2018*; *Whitaker et al., 2016*). In brief, we studied 208 healthy individuals enrolled in an accelerated longitudinal study, with approximately equal numbers of males and females in each of five age-related strata that collectively spanned the time period from 14 to 25 years coinciding with transition from adolescence to young adulthood. Participants (48% female) had a mean age of 18.82 years (range = 14–25 years) at baseline and 19.95 years (15–26 years) at follow-up. The average interval between baseline and follow-up scan was 11.28 months (range = 6–12 months). See Materials and methods for details on participant selection, image processing, and analysis.

### Macroscale structural connectome manifold

For every participant, we built cortex-wide structural connectome manifolds formed by the eigenvectors displaying spatial gradients in structural connectome organization using non-linear dimensionality reduction techniques (*Vos de Wael et al., 2020a*; *Vos de Wael et al., 2020b*, https://github.com/MICA-MNI/BrainSpace). Individual manifolds were aligned to a template manifold estimated from a hold-out dataset (see Materials and methods) (*Langs et al., 2015*; *Vos de Wael et al., 2020a*). Three eigenvectors (E1, E2, and E3) explained approximately 50% of information in the template affinity matrix (i.e., 20.7/15.8/13.5% for E1/E2/E3, respectively), with each eigenvector showing a different axis of spatial variation across the cortical mantle (*Figure 1A*). Eigenvectors depicted a continuous differentiation between medial and lateral cortices (E1), between inferior and superior cortices (E2), and between anterior and posterior areas (E3). For each participant and time point, we calculated *manifold eccentricity*, which depicts how far each node is located from the center of the template manifold (see Materials and methods). It thus quantifies the changes in eigenvectors between the time points in terms of expansion and contraction instead of comparing multidimensional connectome manifolds (*Bethlehem et al., 2020*). The manifold eccentricity showed high values in frontal and somatomotor regions, while temporoparietal, visual, and limbic regions showed low values (*Figure 1B*).

### Changes in manifold eccentricity across age

Leveraging linear mixed effect models that additionally controlled for effects of sex, site, head motion, and subject-specific random intercepts (*Worsley et al., 2009*), we assessed changes in manifold eccentricity across age (see Materials and methods). Manifold eccentricity expanded as age increased, especially in bilateral prefrontal and temporal areas, as well as left early visual and right lateral parietal cortices (false discovery rate [FDR] < 0.05; *Benjamini and Hochberg, 1995*; *Figure 1C*). Stratifying these effects along four cortical hierarchical levels, defined using an established taxonomy based on patterns of laminar differentiation and tract-tracing data in non-human primates (*Mesulam, 1998*), we identified peak effects in heteromodal association and paralimbic areas (*Figure 1D*). Convergent findings were observed when analyzing the effects with respect to intrinsic functional communities (*Yeo et al., 2011*), showing highest effects in default mode and limbic areas followed by visual and frontoparietal cortices. No significant contraction of manifold eccentricity was observed. In addition, we could not find any significant effects when we fitted the model with a quadratic form of age (i.e., $age^2$), indicating the manifold eccentricity linearly increases across age.

To conceptualize the findings derived from manifold eccentricity with respect to conventional network topologies, we correlated manifold eccentricity changes with several graph-theoretical measures of structural connectome (*Figure 1—figure supplement 1*; *Rubinov and Sporns, 2010*). We first defined six spatially contiguous clusters within the regions that showed significant age-related changes in manifold eccentricity (see *Figure 1C*) and correlated within-subject changes in manifold eccentricity with developmental changes in degree centrality, connectivity distance, and modular parameters (i.e., within-module degree and participation coefficient based on modules defined via Louvain's community detection algorithm [*Blondel et al., 2008*]; see Materials and methods;

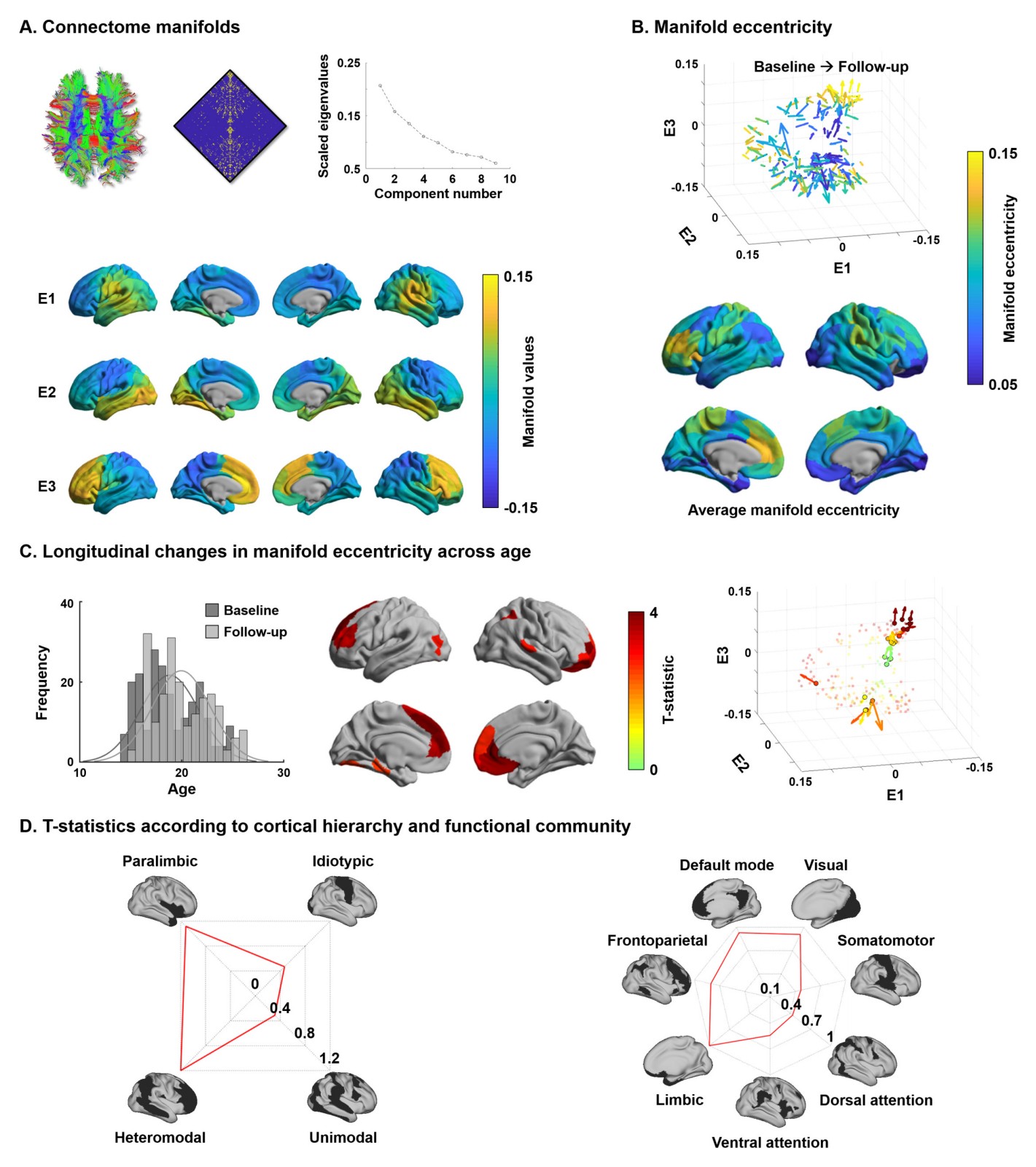

**Figure 1.** Structural connectome manifolds. (**A**) Systematic fiber tracking based on diffusion magnetic resonance imaging generated a cortex-wide structural connectome, which was subjected to diffusion map embedding. As shown in the scree plot, three eigenvectors (E1, E2, E3) accounted for approximately 50% information of connectome data, and each depicted a different gradual transition across the cortical mantle. (**B**) Manifold eccentricity measured by Euclidean distance between the template center and each data point. Arrows depict average positional change in

*Figure 1 continued on next page*

*Figure 1 continued*

connectivity space from baseline to follow-up. The color of each arrow represents each brain region mapped on the surface on the bottom. (C) The histogram represents age distribution of all subjects at baseline and follow-up. The colors on brain surfaces indicate t-statistics of regions showing significant longitudinal changes in manifold eccentricity across age, following multiple comparisons correction with a false discovery rate < 0.05. Datapoint colors in the scatter plot represent t-statistics. Identified regions are represented with arrows that originate from baseline to follow-up. (D) Stratification of age-related changes in manifold eccentricity according to prior models of cortical hierarchy (*Mesulam, 1998*) and functional magnetic resonance imaging communities (*Yeo et al., 2011*).

The online version of this article includes the following source data and figure supplement(s) for figure 1:

**Source data 1.** Source files for connectome manifolds and age-related changes in manifold eccentricity.
**Figure supplement 1.** Association between structural connectome manifold and connectome topology measures.
**Figure supplement 2.** Modular structures.
**Figure supplement 3.** Age-related trends in connectome topology measures.
**Figure supplement 4.** Structural connectome manifolds using Schaefer 300 atlas.
**Figure supplement 5.** Sensitivity analysis for site and sex.
**Figure supplement 6.** Longitudinal changes in manifold eccentricity, after excluding participants with the lowest correspondence to template manifolds.
**Figure supplement 7.** Structural connectome manifolds generated using principal component analysis.
**Figure supplement 8.** Longitudinal changes in graph measures across age.
**Figure supplement 9.** Longitudinal changes in manifold eccentricity calculated using all eigenvectors.
**Figure supplement 10.** Connectome manifolds estimated using group consistency method.
**Figure supplement 11.** Structural connectome manifolds using a structural parcellation.
**Figure supplement 12.** Longitudinal changes in edge weights of structural connectome.
**Figure supplement 13.** Longitudinal changes in manifold eccentricity using a subset of participants who completed Tanner scale.
**Figure supplement 14.** Structural connectome manifolds using different template dataset.
**Figure supplement 15.** A schema of manifold eccentricity for three eigenvectors.

*Figure 1—figure supplement 2*). We found significant positive associations for degree centrality and within-module degree, suggesting that connectome manifold expansion reflects a concurrent increase of overall connectivity, particularly within modules. Stratifying changes in manifold eccentricity, as well as connectome topology measures, according to the discretized age bins confirmed these age-related trends (*Figure 1—figure supplement 3*). Indeed, except for participation coefficient, values in general increased from childhood to young adulthood.

## Effects of cortical morphology and microstructure

Previous studies demonstrated significant changes in cortical morphology and microstructure during adolescence, showing co-occurring reductions in cortical thickness and MT skewness, the latter being an index of depth-dependent intracortical myelin changes in multiple lobes (*Gogtay et al., 2004*; *Khundrakpam et al., 2017*; *Paquola et al., 2019a*; *Shaw et al., 2006*). We replicated these findings by showing cortical thinning in almost all brain regions across the studied age window as well as reductions in depth-dependent MT skewness, suggestive of supragranular enrichment of myelin (*Figure 2A*). To evaluate whether the age-related changes in manifold eccentricity were robust above and beyond these regional changes in cortical thickness and MT, we implemented linear mixed effect models including cortical thickness and MT as covariates in the analysis of developmental change in manifold eccentricity (*Figure 2B*). While we observed virtually identical spatial patterns of manifold eccentricity changes in models that controlled for thickness, MT skewness, and both, age-related effects in regions of significant manifold eccentricity findings (see *Figure 1C*) were reduced in models that additionally controlled for these covariates (average reduction of t-value in models controlling for thickness/MT skewness/both = 42/18/68%).

## Age-related changes in subcortico-cortical connectivity

Besides visualizing these changes in cortico-cortical connectivity, we also capitalized on the manifold representation to assess adolescent changes in the connectivity of subcortical regions, to obtain a more holistic insight into whole-brain connectome reconfigurations during this time period, and to examine whether subcortical connectivity patterns undergo parallel developmental trajectories (*Hwang et al., 2017*; *Shine et al., 2019*). Specifically, we assessed changes in subcortical-weighted manifolds across age, defined by projecting the streamline strength of subcortical regions to cortical

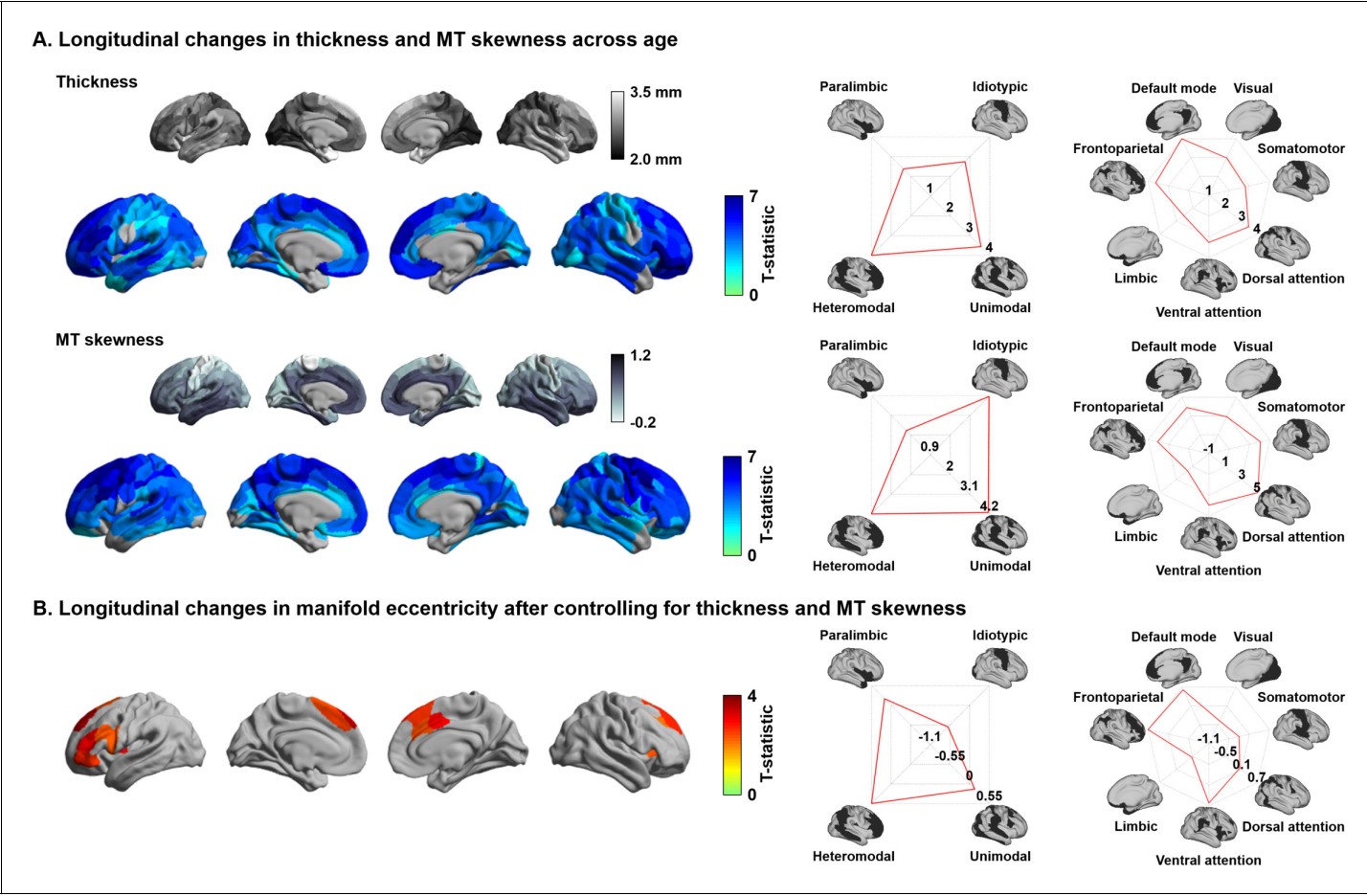

**Figure 2.** Age-related effects on macro- and microstructural metrics of cortical anatomy. (**A**) The t-statistics of identified regions that showed significant age-related changes in cortical thickness (upper row) and magnetization transfer ratio MT (bottom row), and stratification of t-statistics according to cortical hierarchy (*Mesulam, 1998*) and functional community (*Yeo et al., 2011*). (**B**) Age-related changes in manifold eccentricity after controlling for cortical thickness and MT.

The online version of this article includes the following source data for figure 2:

**Source data 1.** Source files for age-related changes in cortical thickness and magnetization transfer ratio.

targets to the manifold space (see Materials and methods). Such an analysis situates changes in sub-cortico-cortical pathways in the macroscale context of cortico-cortical connectivity identified in the previous analyses. After multiple comparisons correction, the caudate and thalamus showed significant age-related effects on subcortical-weighted manifolds (FDR < 0.05; *Figure 3*), and marginal effects were observed in the putamen, pallidum, and hippocampus (FDR < 0.1).

## Transcriptomic association analysis

Connectome organization, in general, and macroscale gradients, in particular, have been argued to reflect genetic expression profiles, underscoring the close link between the physical layout of the brain and innate transcriptional patterning (*Buckner and Krienen, 2013*; *Fornito et al., 2019*). Here, we carried out a transcriptomic association analysis and developmental enrichment analyses to contextualize the age-related manifold eccentricity changes with respect to patterns of *post-mortem* gene expression from a sample of independent adults (*Figure 4A*). Specifically, leveraging mixed effect models, we associated the spatial patterns of manifold change across age in the NSPN sample (controlling for covariation of cortical thickness and MT) with cortical maps of *post-mortem* gene expression data from the Allen Institute for Brain Sciences (*Arnatkeviciute et al., 2019*; *Gorgolewski et al., 2015*; *Gorgolewski et al., 2014*; *Hawrylycz et al., 2012*; *Markello et al.,*

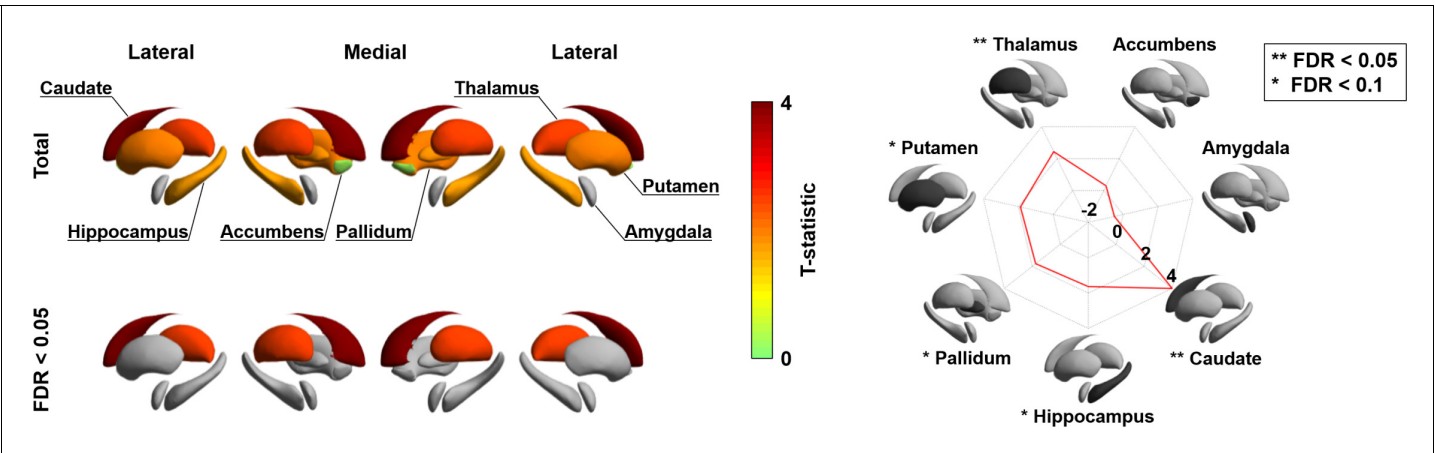

**Figure 3.** Longitudinal changes in subcortical-weighted manifolds. The t-statistics of age-related changes in subcortical-weighted manifolds. The effects of each subcortical region are reported on the radar plot. FDR: false discovery rate.

The online version of this article includes the following source data for figure 3:

**Source data 1.** Source files for age-related changes in subcortical-weighted manifolds.

*2020*). Among the list of most strongly associated genes (FDR < 0.05), we selected only genes that were consistently expressed across different donors (r > 0.5) (*Arnatkeviciute et al., 2019*; *Hawrylycz et al., 2012*; *Markello et al., 2020*; *Supplementary file 1*). We performed developmental gene set enrichment analysis using the cell-type-specific expression analysis (CSEA) tool, which compares the selected gene list with developmental enrichment profiles (see Materials and methods) (*Dougherty et al., 2010*; *Xu et al., 2014*). This analysis highlights developmental time windows across macroscopic brain regions in which genes are strongly expressed. We found marked expression of the genes enriched from childhood onward in the cortex, thalamus, and cerebellum (FDR < 0.001; *Figure 4B*). Although signal was reduced, genes were also enriched for expression in the striatum at the transition from childhood to adolescence (FDR < 0.05). On the other hand, identified genes were not found to be expressed in the hippocampus and amygdala.

## Association between connectome manifold and cognitive function

Finally, to establish associations between connectome reconfigurations and cognitive functioning, we utilized supervised machine learning to predict full IQ at follow-up using manifold eccentricity features. Independent variables were combinations of cortical and subcortical manifold features at baseline and their age-related trajectory data. We used elastic net regularization with nested ten-fold cross-validation (*Cawley and Talbot, 2010*; *Parvandeh et al., 2020*; *Tenenbaum et al., 2000*; *Varma and Simon, 2006*; *Zou and Hastie, 2005*) (see Materials and methods), and repeated the prediction 100 times with different training and test dataset compositions to mitigate subject selection bias. Across cross-validation and iterations, 6.24 ± 5.74 (mean ± SD) features were selected to predict IQ using manifold eccentricity of cortical regions at baseline, 6.20 ± 5.14 cortical features at baseline and maturational change, 5.45 ± 5.99 cortical and subcortical features at baseline, and 5.16 ± 5.43 at baseline and maturational change, suggesting that adding more independent variables may not per se lead to improvement in prediction accuracy. The manifold eccentricity of cortical regions at baseline significantly predicted future IQ score (mean ± SD r = 0.14 ± 0.04; mean absolute error [MAE] = 8.93 ± 0.16, p=0.09). Prediction performance was slightly improved when we combined the manifold eccentricity both at baseline and differences between follow-up and baseline (r = 0.18 ± 0.04; MAE = 9.10 ± 0.19, p=0.04) (*Figure 5A*). Notably, prediction accuracy was improved if we additionally considered subcortical manifold features (baseline: r = 0.17 ± 0.03; MAE = 8.74 ± 0.11, p=0.04; baseline and maturational change: r = 0.21 ± 0.02; MAE = 8.86 ± 0.14, p=0.01) (*Figure 5B*). The regions showing strongest predictive validity for IQ were prefrontal, parietal, and temporal cortices, as well as the caudate and thalamus. The probability map of the selected brain regions (bottom right of *Figure 5B*) was further decoded using Neurosynth (*Yarkoni et al.,*

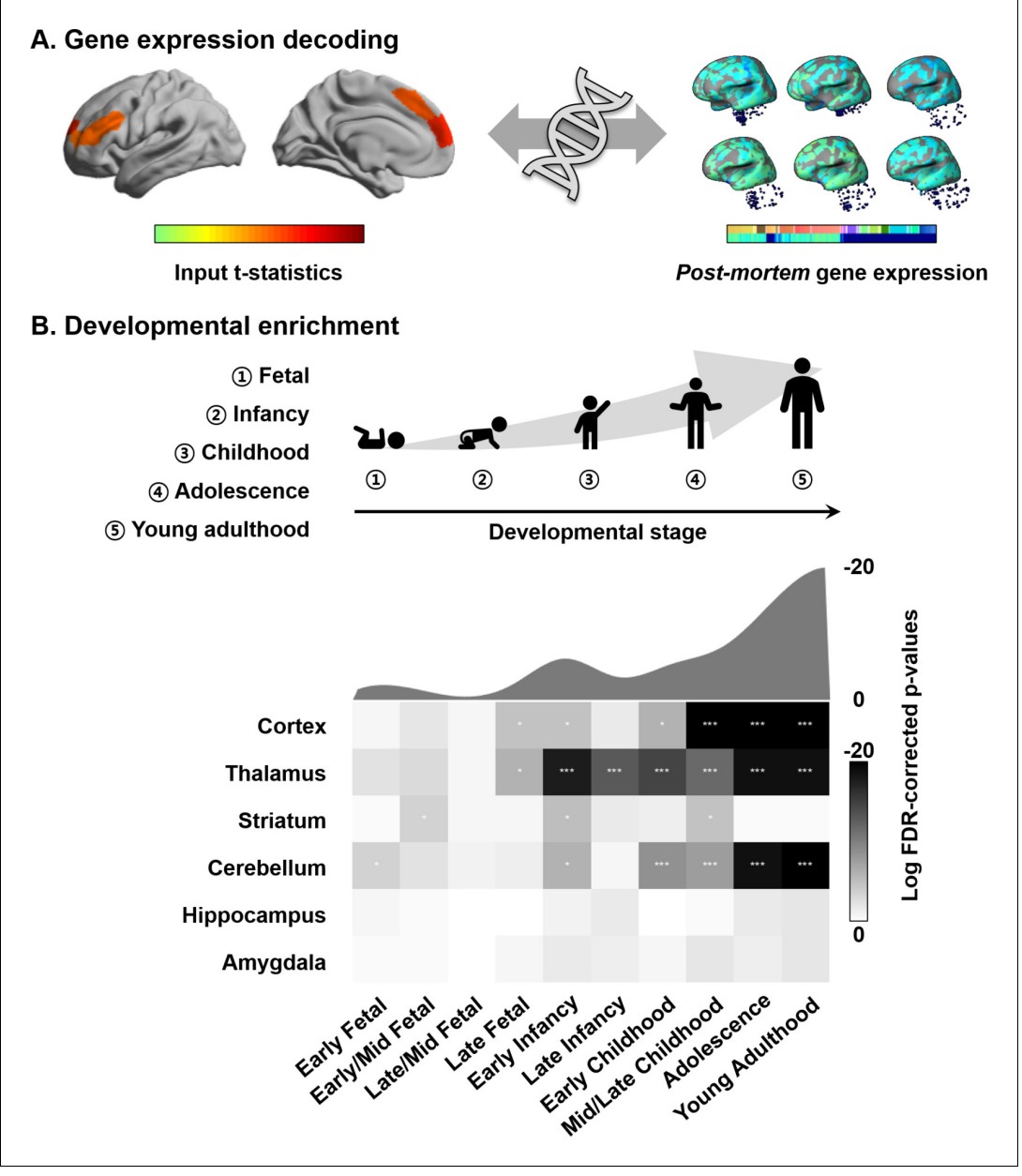

**Figure 4.** Transcriptomic analysis. (**A**) Gene decoding process by associating t-statistics from the linear mixed effect model with *post-mortem* gene expression maps. (**B**) We identified genes that were spatially correlated with the input t-statistic map (false discovery rate [FDR] < 0.05) and selected only those that were furthermore consistently expressed across different donors (r > 0.5). These genes were input to a developmental enrichment analysis, showing strong associations with cortex, thalamus, striatum, and cerebellum during the childhood-to-adulthood time window. The degree of gene expression for developmental windows is reported on the bottom. The curve represents log transformed FDR-corrected p-values, averaged across the brain regions for each of the time windows reported on the bottom. ***FDR < 0.001, ** FDR < 0.01, *FDR < 0.05.

The online version of this article includes the following source data for figure 4:

**Source data 1.** Source files for developmental enrichment profiles.

---

*2011*), revealing strong associations with higher-order cognitive and social terms (*Figure 5—figure supplement 1*). We compared the prediction performance of our model with a baseline model, where IQ of the test set was simple average of training set (r = −0.15 ± 0.06, MAE = 8.98 ± 0.04, p=0.12; see Materials and methods). We found that our model outperformed this baseline model

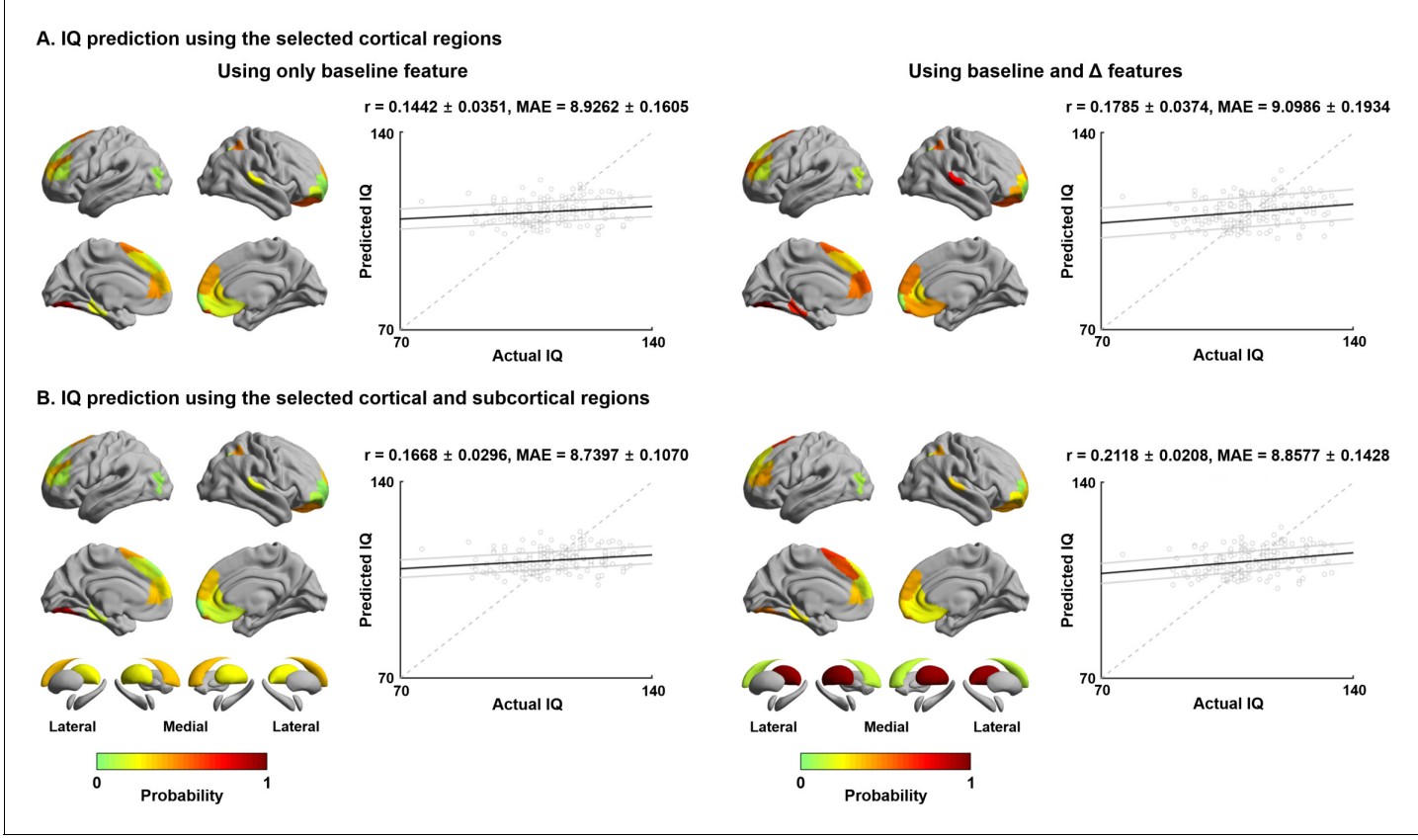

**Figure 5.** Intelligence quotient (IQ) prediction by baseline and follow-up measures of cortical and subcortical manifolds. (A) Probability of selected brain regions across ten-fold cross-validation and 100 repetitions for predicting future IQ using only baseline manifold eccentricity (left), and both baseline and maturational change in the feature (right). Correlations between actual and predicted IQ are reported. Black lines indicate mean correlation, and gray lines represent 95% confidence interval for 100 iterations with different training/test dataset. (B) The prediction performance when both cortical and subcortical features were considered. MAE: mean absolute error.

The online version of this article includes the following source data and figure supplement(s) for figure 5:

**Source data 1.** Source files for selected probability as well as actual and predicted intelligence quotient.
**Figure supplement 1.** Cognitive decoding of the selected regions for intelligence quotient (IQ) prediction.
**Figure supplement 2.** Intelligence quotient (IQ) prediction using regression tree approach.

(Meng's z-test p < 0.001) (*Meng et al., 1992*). We also predicted the change of IQ between the baseline and follow-up, instead of IQ at follow-up, using the imaging features. However, we could not find significant results.

## Sensitivity analysis
### Spatial scale
Repeating the longitudinal modeling with a different spatial scale (i.e., 300 parcels), findings were highly consistent (*Figure 1—figure supplement 4*).

### Site and sex effects
Furthermore, manifold eccentricity of the identified cortical regions and age consistently correlated positively across different sites and within both biological sexes, yielding non-significant interaction effects (*Figure 1—figure supplement 5*).

## Different parameters for diffusion map embedding

When we changed parameters of diffusion map embedding for generating connectome manifolds (see Materials and methods), t-statistic maps of age-related changes in manifold eccentricity were largely consistent (mean ± SD linear correlation r = 0.92 ± 0.10).

## Gradient alignment fidelity

When calculating linear correlations between template and individual manifolds before and after alignment, we found significant increases after alignment (r = 0.92 ± 0.03/0.93 ± 0.03/0.94 ± 0.03) compared to before alignment (−0.02 ± 0.03/−0.001 ± 0.37/0.003 ± 0.12) for E1/E2/E3, respectively, supporting effectiveness of alignment. After excluding 10% of subjects with poor alignment (cutoff r = 0.83; the new set was correlated with the template manifold, r = 0.94 ± 0.01), we found consistent age-related changes in manifold eccentricity (*Figure 1—figure supplement 6*), with the t-statistic map showing strong correlation to the map derived in the whole sample (r = 0.97, p<0.001).

## Connectome manifold generation using principal component analysis

In a separate analysis, we generated eigenvectors using principal component analysis (*Wold et al., 1987*), instead of diffusion map embedding (*Coifman and Lafon, 2006*), and found consistent spatial maps (linear correlation = 0.998 ± 0.001 across E1/E2/E3; *Figure 1—figure supplement 7A*) and longitudinal findings (*Figure 1—figure supplement 7B*).

## Longitudinal changes in graph-theoretical measures

Repeating the longitudinal modeling using graph-theoretical centrality measures, we found significant age-related longitudinal changes in degree and eigenvector centrality, while betweenness centrality did not reveal significant effects, in similar regions to those that had significant age-related changes in manifold eccentricity (*Figure 1—figure supplement 8*). Correlating the effect size maps for manifold eccentricity and each graph measure, we found a significant yet variable spatial similarity of the effect maps (betweenness centrality: r = 0.18, spin-test p = 0.02; degree centrality: r = 0.57, p < 0.001; eigenvector centrality: r = 0.47, p < 0.001).

## Manifold eccentricity based on all eigenvectors

Repeating manifold eccentricity calculation and age modeling using all eigenvectors, instead of using only the first three, we observed relatively consistent results with our original findings (linear correlation of manifold eccentricity r = 0.54, p<0.001; t-statistic map r = 0.68, p<0.001), also pointing to manifold expansion in transmodal cortices (*Figure 1—figure supplement 9*).

## Robustness of group representative structural connectome

We compared gradients derived from the group representative structural connectome, based on (i) distance-dependent thresholding (*Betzel et al., 2019*) and (ii) consistency thresholding (*Wang et al., 2019*; *Figure 1—figure supplement 10*). We found high similarity in spatial maps of the estimated manifolds (r = 0.89 ± 0.01 for E1; 0.93 ± 0.004 for E2; 0.85 ± 0.01 for E3 across six different thresholds), indicating robustness.

## Connectome manifolds based on structural parcellation

We repeated our analyses with a structural parcellation, defined using a sub-parcellation of folding based on the Desikan–Killiany atlas (*Desikan et al., 2006*; *Vos de Wael et al., 2020a*; *Figure 1—figure supplement 11*). Despite slight differences in the topography of manifold eccentricity in lateral prefrontal, temporal, and occipital cortices, we could replicate strong age-related effects in heteromodal association areas, together with effects in caudate and hippocampus (FDR < 0.05), and marginally in thalamus (FDR < 0.1).

## Longitudinal modeling using edge weights

Repeating the longitudinal modeling across age using connectome edge weights, we found significant increases in edge weights within frontoparietal and default mode networks, as well as in

attention and sensory networks (FDR < 0.05; *Figure 1—figure supplement 12*), consistent with findings based on manifold eccentricity.

### Manifold eccentricity and pubertal stages

We repeated the longitudinal modeling within a subset of participants who completed the Tanner scale (n = 73) (*Marshall and Tanner, 1970*; *Marshall and Tanner, 1969*) and found relatively consistent albeit weaker age-related changes in manifold eccentricity as for the overall sample (*Figure 1—figure supplement 13A*). Notably, manifold eccentricity within the identified regions derived from overall sample and Tanner scale revealed a significant interaction effect (t = 2.36, p=0.01; *Figure 1—figure supplement 13B*), suggesting that participants in early pubertal stages show more marked changes in manifold eccentricity across age compared to those in later stages.

### IQ prediction using nonlinear model

We predicted IQ at follow-up using a regression tree method (*Breiman et al., 1984*), instead of linear regression model, but we could not find improved prediction performance (*Figure 5—figure supplement 2*).

## Discussion

The current study tracked whole-brain structural connectome maturation from adolescence to young adulthood in an accelerated longitudinal imaging cohort (*Kiddle et al., 2018*; *Whitaker et al., 2016*). Capitalizing on advanced manifold learning techniques applied to dMRI-derived connectomes, we established that higher-order association cortices in prefrontal, medial and superior temporal areas, as well as parieto-occipital regions, expanded in their connectome manifold representation indicative of an increased differentiation of these systems from the rest of the brain in adolescence. Parallel topological analysis based on graph theory indicated that these changes anatomically coincided with increases in the within-module connectivity of transmodal cortices. Findings were consistent across the different acquisition sites and biological sexes, and similar albeit slightly weaker when correcting connectivity manifolds for MRI-based measures of macrostructure (cortical thickness) and microstructure (skewness of MT depth profile). In addition to the cortical manifold expansion, we found parallel reconfigurations of subcortical connectivity patterns for the caudate and thalamus. Decoding our findings with *post-mortem* gene expression maps implicated genes enriched in adolescence and young adulthood, again pointing to both cortical as well as subcortical targets. Finally, the combination of both cortical and subcortical manifold measures predicted behavioral measures of intelligence at follow-up, with higher performance than cortical or subcortical data alone. Collectively, our findings provide new insights into adolescent structural connectome maturation and indicate how multiple scales of cortical and subcortical organization can interact in typical neurodevelopment.

Leveraging advanced manifold learning, we depicted macroscale connectome organization along continuous cortical axes. Similar approaches have previously been harnessed to decompose microstructural (*Paquola et al., 2019b*; *Paquola et al., 2019a*) and functional MRI (*Bethlehem et al., 2020*; *Hong et al., 2019*; *Margulies et al., 2016*; *Murphy et al., 2019*; *Vos de Wael et al., 2020a*). These techniques are appealing as they offer a low-dimensional perspective on connectome reconfigurations in a data-driven and spatially unconstrained manner. In our longitudinal study, we could identify marked connectome expansion during adolescence, mainly encompassing transmodal and heteromodal association cortex in prefrontal, temporal, and posterior regions, the territories known to mature later in development (*Gogtay et al., 2004*; *Shaw et al., 2006*). Findings remained consistent when we considered a linear dimensionality reduction technique, suggesting robustness to methodological details of this analysis. Connectome expansion can be understood as an overall greater differentiation of the connectivity of these areas from the rest of the brain as they would then cover wider portions of the corresponding manifold space. Manifold expansion in higher-order areas correlated with an increase in their within-module connectivity, but not with participation coefficient and connectivity distance measures that would be more reflective of their between-module connectivity. In light of potential limitations of dMRI tractography in detecting long-distance fiber tracts (*Betzel et al., 2019*; *Maier-Hein et al., 2017*), we cannot rule out a reduced sensitivity of our

approach for the study of long-range inter-regional connections. Nevertheless, our diffusion modeling was based on constrained spherical-deconvolution approaches together with SIFT2-based tractogram filtering, in addition to using a distance-dependent thresholding approach that may have partially mitigated these limitations (*Betzel et al., 2019*). Thus, our findings do overall confirm and extend prior dMRI studies that have focused on specific tracts and that have indicated considerable developmental shifts in diffusion parameters, such as increases in fractional anisotropy and decreases in mean diffusivity in early and late adolescence (*Olson et al., 2009*). Other studies have furthermore reported increased streamline count estimates (*Genc et al., 2020*). In this context, our macroscale manifold findings likely reflect an ongoing consolidation of transmodal cortical communities. These findings align with prior graph-theoretical studies, which have pointed to concurrent increases in network integration and consolidation of network hubs from late childhood to early adulthood (*Baker et al., 2015*; *Lebel and Beaulieu, 2011*; *Oldham and Fornito, 2019*). Considering their distributed regional substrate, these network effects are likely driven by the ongoing maturation of fiber bundles that interconnect these higher-order cortices, including superior longitudinal fascicules, but also thalamic and basal ganglia pathways (*Tamnes et al., 2010*), throughout adolescence.

Projecting manifold solutions back onto cortical surfaces allowed us to integrate our connectome manifold results with morphometric and intracortical intensity indices obtained via structural and quantitative MRI contrasts in the same participants. We were thus able to balance the network-level effects against trajectories of intracortical remodeling. Longitudinal changes in these cortical features were overall in agreement with prior work, suggesting marked reductions in cortical thickness in adolescence (*Khundrakpam et al., 2017*; *Shaw et al., 2006*), possibly reflecting synaptic pruning processes (*Petanjek et al., 2011*) together with decreases in the skewness of intracortical MT profiles, a feature sensitive to preferential myelination of supragranular layers (*Paquola et al., 2019a*). Although we still observed significant age-related changes in manifold eccentricity after controlling for these intracortical and morphological measures, the effect sizes of our findings were reduced. This effect was particularly evident when running a parallel analysis that additionally controlled for depth-dependent shifts in cortical microstructure, a finding in line with more generally demonstrated links between cortical microstructural depth profiles and inter-regional connectivity (*Paquola et al., 2019b*). In the context of adolescence and prior findings in the NSPN dataset (*Paquola et al., 2019a*), these results thus suggest a coupled process that affects depth-dependent shifts in cortical myeloarchitecture, on the one hand, and adolescent shifts in macroscale connectome organization, on the other hand, as shown by our longitudinal manifold analyses.

In addition to emphasizing a distributed set of association cortices and their cortico-cortical connections, analysis of subcortico-cortical connectivity patterns highlighted parallel developmental processes in several subcortical structures and their connections, particularly the caudate and thalamus. These findings were independently supported by transcriptomic association studies and developmental enrichment analyses, which implicated genes expressed in cortical regions and these subcortical structures during late childhood, adolescence, and young adulthood. The caudate nucleus of the striatum has long been recognized to play an important role in mediating large-scale cortical network organization (*Aglioti, 1997*; *Alexander and Crutcher, 1990*; *Graybiel, 1995*), a finding also increasingly recognized in the connectome literature (*Hwang et al., 2017*; *Müller et al., 2020*; *Shine et al., 2019*). It is known to modulate activity in prefrontal association areas during memory-driven internal thought processes (*Aglioti, 1997*), and higher-order cognitive functions, notably motivational processes, decision making, as well as cognitive control and executive functions more generally (*Aglioti, 1997*; *Graybiel, 1995*). Regions of the striatum participate in dense cortico-subcortical feedback loops and exchange neural activity through dense connections with adjacent basal ganglia structures as well as the thalamus (*Aglioti, 1997*; *Alexander and Crutcher, 1990*; *Shine, 2021*). Associating macroscopic changes in manifold eccentricity with *post-mortem* microarray data provided by the Allen Human Brain Atlas (*Arnatkeviciute et al., 2019*; *Fornito et al., 2019*; *Gorgolewski et al., 2014*; *Hawrylycz et al., 2015*; *Thompson et al., 2013*), we identified gene sets expressed in cortical regions and subcortical structures of the thalamus and striatum during late childhood, adolescence, and young adulthood. Despite these findings being associative and based on separate datasets, they overall support our results that brain network maturation from late childhood until early adulthood implicates micro- and macroscale factors in both subcortical and cortical networks. Coupled network and molecular changes may ultimately change subcortical and cortical circuit properties, including the balance of excitation and inhibition (E/I). Human brain

development involves spatio-temporal waves of gene expression changes across different brain regions and developmental time windows (*Ip et al., 2010*; *Kang et al., 2011*; *Shin et al., 2018*). In the study of adolescent development, prior studies have suggested shifts in E/I balance, evolving from a dominant inhibitory bias in early developmental stages towards stronger excitatory drive in later stages, and suggested that these may underlie the maturation of cognitive functions such as working memory and executive control (*Dorrn et al., 2010*; *Lander et al., 2017*; *Liu et al., 2007*). In common neurodevelopmental disorders, including autism, schizophrenia, and attention-deficit hyperactivity disorder, imbalances in cortical E/I and cortico-subcortical network function have been demonstrated (*Cellot and Cherubini, 2014*; *Gandal et al., 2018*; *Lee et al., 2017*; *Lewis et al., 2005*; *Nelson and Valakh, 2015*; *Park et al., 2021a*; *Sohal and Rubenstein, 2019*; *Trakoshis et al., 2020*), potentially downstream to perturbations of different neurotransmitter systems, such as inter-neuron-mediated GABA transmission (*Bonaventura et al., 2017*; *Kilb, 2012*; *Liu et al., 2007*; *Park et al., 2021a*; *Silveri et al., 2013*; *Trakoshis et al., 2020*; *Tziortzi et al., 2014*).

Higher-order cognitive function implicates functionally relevant whole-brain network mechanisms, and its prediction may thus leverage structurally governed principles of network integration and segregation. Application of a supervised machine learning framework with cross-validation and regularization to our cohort demonstrated that it is possible to predict inter-individual variations in future IQ from structural connectome manifold data. These findings complement conceptual accounts linking brain organization to cognitive function (*Margulies et al., 2016*; *Mesulam, 1998*) and earlier efforts to predict IQ measures from inter-regional measures of connectivity and graph-theoretical indices of network topology (*Greene et al., 2018*). Notably, evaluations of several feature combinations highlighted that predictive performance was highest when including both baseline and trajectory data, and when complementing cortical and subcortical manifold features. These findings re-emphasize the benefits of incorporating subcortical nodes in the characterization of large-scale cortical network organization and overall cognitive function (*Alves et al., 2019*; *Müller et al., 2020*; *Shine, 2021*; *Shine et al., 2019*). Of note, although our model significantly outperformed a baseline model, the relationship between the actual and predicted IQ scores did not locate on the equality line and the strength of the association was rather weak. Further improvements in brain-based IQ prediction in adolescence, for example, through combinations of structural and functional imaging features, will be a focus of future work.

Adolescence is a time characterized by ongoing brain changes (*Baum et al., 2020*; *Gogtay et al., 2004*; *Larsen and Luna, 2018*; *Menon, 2013*; *Shaw et al., 2006*), gradually increasing independence from caregivers, accompanied by strong increments in knowledge and our ability to think more abstractly and to cooperate with our peers to achieve common goals. On the other hand, adolescence is also a sensitive time window for risk taking, the development of addictions, and is associated with high rates of onset of several psychiatric disorders (*Hong et al., 2019*; *Khundrakpam et al., 2017*). Our study has shown that structural brain network organization continues to mature significantly during this time period, with higher-order association cortices in prefrontal and posterior regions especially showing an expansion of their corresponding connectome manifold signature. Findings were related to an increased strengthening of intra-community connectivity as well as cortico-subcortical connectivity to thalamo-striatal regions. Although the current work was restricted to a longitudinal sample of typically developing adolescents, our framework may be useful to explore multiscale network perturbations in cohorts with a psychiatric diagnosis or those at risk for addiction or atypical neurodevelopment.

## Materials and methods

### Participants

We obtained imaging and phenotypic data from the NSPN 2400 cohort, which contains questionnaire data on 2402 individuals (with MRI data on a subset of ~300) from adolescence to young adulthood in a longitudinal setting (*Kiddle et al., 2018*; *Whitaker et al., 2016*). The NSPN study was ethically approved by the National Research Ethics Service and conducted in accordance with NHS research governance standards. All participants provided informed consent in writing, with additional parental consent for participants aged less than 16 years at enrollment. Included participants completed quality-controlled (see Data preprocessing section) multimodal MRI scans consisting of

T1-weighted, MT, and dMRI for at least two time points. Our final sample consisted of a total of 208 participants (48% female; mean [range] age = 18.82 [14–25] years at baseline and 19.95 [15–26] at follow-up; inter-scan interval of 11.28 [6–12] months), collected from three different UK sites: Wolfson Brain Imaging Centre and MRC Cognition and Brain Sciences Unit in Cambridge; and University College London. We divided the participants into template and non-template cohorts with matched age, sex, and site ratio. The template dataset (n = 30; 50% female; mean [range] age = 18.69 [15–24] years at baseline and 19.84 ± 2.66 [16–25] at follow-up) was used for constructing the group mean template manifold and the non-template dataset (n = 178; 47% female; mean [range] age = 18.84 [14–25] years at baseline and 19.97 [15–26] at follow-up) was used for conducting main analyses. Of note, changing the template dataset composition did not markedly affect main findings (*Figure 1—figure supplement 14*).

## MRI acquisition

Imaging data were obtained using Siemens Magnetom TIM Trio 3T scanners. T1-weighted and MT sequences were acquired using a quantitative multiparameter mapping sequence (repetition time [TR]/flip angle = 18.7 ms/20° for T1-weighted and 23.7 ms/6° for MT; six equidistance echo times [TE] = 2.2–14.7 ms; voxel size = 1 mm$^3$; 176 slices; field of view [FOV] = 256 × 240 mm; matrix size = 256 × 240 × 176) (*Weiskopf et al., 2013*). The dMRI data were acquired using a spin-echo echo-planar imaging sequence (TR = 8700 ms; TE = 90 ms; flip angle = 90°; voxel size = 2 mm$^3$; 70 slices; FOV = 192 × 192 mm$^2$; matrix size = 96 × 96 × 70; b-value = 1000 s/mm$^2$; 63 diffusion directions; and 6 b0 images).

## Data preprocessing

T1-weighted data were processed using the fusion of neuroimaging preprocessing (FuNP) pipeline integrating AFNI, FSL, FreeSurfer, ANTs, and Workbench (*Avants et al., 2011*; *Cox, 1996*; *Fischl, 2012*; *Glasser et al., 2013*; *Jenkinson et al., 2012*; *Park et al., 2019*), which is similar to the minimal preprocessing pipeline for the Human Connectome Project (*Glasser et al., 2013*). Gradient nonlinearity and b0 distortion correction, non-brain tissue removal, and intensity normalization were performed. The white matter and pial surfaces were generated by following the boundaries between different tissues (*Dale et al., 1999*), and they were averaged to generate the midthickness contour, which was used to generate the inflated surface. The spherical surface was registered to the Conte69 template with 164k vertices (*Van Essen et al., 2012*) and downsampled to a 32k vertex mesh. Quality control involved visual inspection of surface reconstruction of T1-weighted data, and cases with faulty cortical segmentation were excluded. Surface-based co-registration between T1-weighted and MT weighted scans was performed. We generated 14 equivolumetric cortical surfaces within the cortex and sampled MT intensity along these surfaces (*Paquola et al., 2019a*). The vertex-wise MT profiles for each surface depth were averaged based on the Schaefer atlas with 200 parcels (*Schaefer et al., 2018*). The dMRI data were processed using MRtrix3 (*Tournier et al., 2019*), including correction for susceptibility distortions, head motion, and eddy currents. We visually inspected the quality of co-registration between the adolescence data and adult-driven surface template as well as parcellation atlas, and all data showed reasonable registration results.

## Structural connectome manifold identification

Structural connectomes were generated from preprocessed dMRI data (*Tournier et al., 2019*). Anatomically constrained tractography was performed using different tissue types derived from the T1-weighted image, including cortical and subcortical gray matter, white matter, and cerebrospinal fluid (*Smith et al., 2012*). Multi-shell and multi-tissue response functions were estimated (*Christiaens et al., 2015*), and constrained spherical-deconvolution and intensity normalization were performed (*Jeurissen et al., 2014*). The tractogram was generated with 40 million streamlines, with a maximum tract length of 250 and a fractional anisotropy cutoff of 0.06. Subsequently, spherical-deconvolution informed filtering of tractograms (SIFT2) was applied to reconstruct whole-brain streamlines weighted by the cross-section multipliers, which considers the fiber bundle's total intra-axonal space across its full cross-sectional extent (*Smith et al., 2015*). The structural connectome was built by mapping the reconstructed cross-section streamlines onto the Schaefer 7-network based atlas with 200 parcels (*Schaefer et al., 2018*) then log-transformed to adjust for the scale

(*Fornito et al., 2016*). We opted for this atlas as it (i) allows contextualization of our findings within macroscale intrinsic functional communities (*Yeo et al., 2011*), (ii) incorporates the option to assess results across different granularities, and (iii) aligns the current study with previous work from our group (*Benkarim et al., 2020*; *Paquola et al., 2020*; *Park et al., 2021b*; *Park et al., 2021a*; *Rodrí-guez-Cruces et al., 2020*) and others (*Baum et al., 2020*; *Betzel et al., 2019*; *Osmanlıoğlu et al., 2019*).

Cortex-wide structural connectome manifolds were identified using BrainSpace (https://github.com/MICA-MNI/BrainSpace; *Vos de Wael et al., 2020a*). First, a template manifold was estimated using a group representative structural connectome of the template dataset. The group representative structural connectome was defined using a distance-dependent thresholding that preserves long-range connections (*Betzel et al., 2019*). An affinity matrix was constructed with a normalized angle kernel, and eigenvectors were estimated via diffusion map embedding (*Figure 1A*), a nonlinear dimensionality reduction technique (*Coifman and Lafon, 2006*) that projects connectome features into low-dimensional manifolds (*Margulies et al., 2016*). This technique is only controlled by a few parameters, computationally efficient, and relatively robust to noise compared to other nonlinear techniques (*Errity and McKenna, 2007*; *Gallos et al., 2020*; *Hong et al., 2020*; *Tenenbaum et al., 2000*), and has been extensively used in the previous gradient mapping literature (*Hong et al., 2019*; *Hong et al., 2020*; *Huntenburg et al., 2017*; *Larivière et al., 2020a*; *Margulies et al., 2016*; *Müller et al., 2020*; *Paquola et al., 2019a*; *Park et al., 2021b*; *Valk et al., 2020*; *Vos de Wael et al., 2020a*). It is controlled by two parameters α and *t*, where α controls the influence of the density of sampling points on the manifold (α = 0, maximal influence; α = 1, no influence) and *t* controls the scale of eigenvalues of the diffusion operator. We set α = 0.5 and t = 0 to retain the global relations between data points in the embedded space, following prior applications (*Hong et al., 2019*; *Margulies et al., 2016*; *Paquola et al., 2019a*; *Paquola et al., 2019b*; *Vos de Wael et al., 2020a*). Briefly, the eigenvectors estimated from the decomposition technique generate a connectivity coordinate system (*Bijsterbosch et al., 2020*; *Haak et al., 2018*; *Huntenburg et al., 2018*; *Margulies et al., 2016*; *Mars et al., 2018*) – the diffusion map, where Euclidean distances in the manifold correspond to diffusion times between the nodes of the network (*Coifman and Lafon, 2006*). In this manifold space, interconnected brain regions with similar connectivity patterns are closely located, and regions with weak similarity in connectivity patterns are located farther apart. After generating the template manifold, individual-level manifolds were estimated from the non-template dataset and aligned to the template manifold via Procrustes alignment (*Langs et al., 2015*; *Vos de Wael et al., 2020a*). To analyze change in the low-dimensional manifold space, we simplified the multivariate eigenvectors into a single scalar value that is., manifold eccentricity (*Figure 1B*). Manifold eccentricity was calculated as the Euclidean distance between the manifold origin and all data points (i.e., brain regions) in manifold space. The template center was defined as the centroid of the first three eigenvectors, which explained 50% variance. Specifically, manifold eccentricity was defined as follows:

$$C_T = \frac{1}{N} \left[ \sum_{i=1}^{N} T(E1)_i, \ \sum_{i=1}^{N} T(E2)_i, \ \sum_{i=1}^{N} T(E3)_i \right] \tag{1}$$

$$ME = \sqrt{\sum_{e=1}^{3} \{I(E_e) - C_T(e)\}^2} \tag{2}$$

$C_T$ is the template manifold origin, $N$ the number of brain regions, $T(\cdot)$ the template manifold, $ME$ the manifold eccentricity, $I(\cdot)$ the individual manifold, and $C_T(e)$ the origin of e-th template manifold. Simply, as shown in *Figure 1—figure supplement 15*, each brain region (i.e., each dot in the scatter plot) is described as a vector from the manifold origin (i.e., triangular mark in the scatter plot), and manifold eccentricity is simply a length (i.e., Euclidean distance) of that vector. Shifts in connectivity patterns of a given region thus will lead to shifts in the vectors, which in turn changes the manifold eccentricity. Thus, manifold eccentricity quantifies global brain organization based in the connectivity space.

## Age-related changes in structural manifolds

We assessed changes in manifold eccentricity across age using a linear mixed effect model (*Worsley et al., 2009*), controlling for effects of sex, site, head motion, and subject-specific random intercept to improve model fit in accelerated longitudinal designs. The t-statistics of each brain region were computed, and we corrected for multiple comparisons by using an FDR threshold of q < 0.05 (*Figure 1C*; *Benjamini and Hochberg, 1995*). We stratified age-related effects based on a seminal model of neural organization and laminar differentiation that contains four cortical hierarchical levels (*Mesulam, 1998*), as well as seven intrinsic functional communities (*Yeo et al., 2011*; *Figure 1D*). To assess the effects with respect to age[2], we repeated implementing a linear mixed effect model by adding a quadratic term of age to the model.

To provide the underlying structure of manifold eccentricity, we compared the changes in manifold eccentricity with those in connectome topology measures. We first defined clusters within the identified regions based on their spatial boundaries (*Figure 1—figure supplement 1A*). Then, we calculated degree centrality, as well as modular measures of within-module degree and participation coefficient using the Brain Connectivity Toolbox (https://sites.google.com/site/bctnet/) (*Rubinov and Sporns, 2010*) and connectivity distance using a recently published approach (*Larivière et al., 2020b*). Degree centrality is defined as the row-wise sum of the weighted connectivity matrix, representing the connection strength of a given node (*Rubinov and Sporns, 2010*). Connectivity distance is a given brain region's geodesic distance to its structurally connected brain areas within the cortex (*Oligschläger et al., 2019*), and it is defined as the multiplication between the geodesic distance and the binarized structural connectome (*Hong et al., 2019*; *Oligschläger et al., 2019*). Within-module degree and participation coefficient are nodal measures reflecting different facets of community organization (*Rubinov and Sporns, 2010*). For each individual subject, community structure was defined using Louvain's algorithm (*Blondel et al., 2008*) and a consistency matrix was constructed, where each element of the matrix represents whether the two different nodes are involved in the same community (i.e., 1) or not (i.e., 0) (*Figure 1—figure supplement 2A*). We constructed the group-wise consistency matrix by averaging the consistency matrix of all subjects and applied k-means clustering (*Figure 1—figure supplement 2B*). The optimal number of clusters was determined using the silhouette coefficient, that is, the k that maximized the silhouette coefficient (*Kannan et al., 2010*). We calculated within-module degree and participation coefficient based on these modules. Within-module degree is the degree centrality within a community, indicating intra-community connection strength, while participation coefficient represents inter-community connectivity (*Rubinov and Sporns, 2010*). We calculated linear correlations between changes in manifold eccentricity and those in each graph-theoretical measure for each cluster (*Figure 1—figure supplement 1B*). The significance of the correlation was corrected using 1000 permutation tests by randomly shuffling subject indices in one of the data vectors, and we corrected for multiple comparisons across clusters using an FDR procedure (*Benjamini and Hochberg, 1995*). To visualize age-related changes in these parameters, we stratified each measure according to discretized age bins (<17, 17–19, 19–21, 21–23, ≥23; *Figure 1—figure supplement 3*).

## Cortical morphology and microstructure

It has been shown that macroscale cortical morphology and microstructure significantly change during development (*Gogtay et al., 2004*; *Khundrakpam et al., 2017*; *Paquola et al., 2019a*; *Shaw et al., 2006*). Here, we confirmed these changes by assessing age-related changes in MRI-based cortical thickness measures and intracortical measures of MT, an index sensitive to myelin content (*Weiskopf et al., 2013*), using linear mixed effect models (*Figure 2A*; *Worsley et al., 2009*). We further regressed out cortical thickness and MT from the connectome manifold eccentricity metric. We then implemented linear mixed effect models using the residuals of manifold measures to assess whether age-related connectome manifold effects exist above and beyond age-related effects on cortical morphology and microstructure (*Figure 2B*).

## Subcortico-cortical connectivity

To assess age-related changes in subcortical manifold organizations in addition to cortical manifold structures, we first parcellated the accumbens, amygdala, caudate, hippocampus, pallidum, putamen, and thalamus for each individual (*Patenaude et al., 2011*), and approximated cross-sectionl

streamlines connect each subcortical region to the rest of the brain. For each individual and each subcortical region, we projected the streamline strength to cortical manifold space by weighting the cortical manifolds with the streamline strength of the connection between each subcortical region and cortical parcels, yielding a matrix with the form of (number of brain regions × number of cortical manifolds). We averaged the matrix across the axis of cortical manifolds to construct subcortical-weighted manifold vector. We assessed age-related changes in the subcortical-weighted manifold using a linear mixed effect model (*Worsley et al., 2009*), controlling for sex, site, head motion, and subject-specific random intercept, and FDR corrected for multiple comparisons (*Figure 3*; *Benjamini and Hochberg, 1995*).

## Transcriptomic analysis

We performed spatial correlation analysis to *post-mortem* gene expression data and carried out a developmental enrichment analysis (*Figure 4*). In brief, we first correlated the t-statistics map, which represents age-related changes in manifold eccentricity that controlled for cortical morphology and microstructure, with the *post-mortem* gene expression maps provided by the Allen Institute using the Neurovault gene decoding tool (*Gorgolewski et al., 2015*; *Gorgolewski et al., 2014*; *Hawrylycz et al., 2012*). Leveraging mixed effect models to associate the input t-statistic map with the genes of six donor brains, Neurovault yields the gene symbols associated with the input spatial map. Gene symbols that passed for a significance level of FDR < 0.05 were further tested whether they are consistently expressed across different donors using abagen toolbox (*Markello et al., 2020*; copy archived at https://github.com/rmarkello/abagen; *Arnatkeviciute et al., 2019*; *Hawrylycz et al., 2012*). For each gene, we estimated whole-brain gene expression map and correlated it between all pairs of donors. Leveraging CSEA developmental expression tool (http://genetics.wustl.edu/jdlab/csea-tool-2; *Dougherty et al., 2010*; *Xu et al., 2014*), we evaluated the significance of overlap between the genes showing consistent whole-brain expression pattern across donors (FDR < 0.05) with RNAseq data obtained from BrainSpan dataset (http://www.brainspan.org). The significance was calculated based on Fisher's exact test (*Fisher, 1922*) with FDR correction (*Benjamini and Hochberg, 1995*). The CSEA tool provides simplified results of gene enrichment profiles along six major brain regions (i.e., cortex, thalamus, striatum, cerebellum, hippocampus, amygdala) across 10 developmental periods (from early fetal to young adulthood) approximated from mouse data, yielding a total of 60 combinations of developmental enrichment profiles (*Xu et al., 2014*). We repeated developmental enrichment analysis using the genes identified from the rotated maps of the age-related changes in manifold eccentricity (100 spherical rotations). For each iteration, we obtained developmental expression profiles using the identified genes, where the FDR-corrected p-values built a null distribution. For each brain division and developmental period, if the actual p-value is placed outside 95% of the null distribution, it was deemed significant. As the Allen Brain Institute repository is composed of adult *post-mortem* datasets, it should be noted that the associated gene symbols represent indirect associations with the input t-statistic map derived from the developmental data.

## Association with the development of cognitive function

Leveraging a supervised machine learning with ten-fold cross-validation, we predicted full IQ score measured by the Wechsler Abbreviated Scale of Intelligence (*Wechsler, 1999*) at follow-up using cortical and subcortical features. Four different feature sets were evaluated: (i) manifold eccentricity of the identified cortical regions at baseline and (ii) manifold eccentricity at baseline and its longitudinal change (i.e., differences between follow-up and baseline), and (iii) cortical manifold eccentricity and subcortical-weighted manifold of the identified regions at baseline and (iv) manifold eccentricity and subcortical-weighted manifold at baseline and their longitudinal changes. For each evaluation, a subset of features that could predict future IQ was identified using elastic net regularization (ρ=0.5) with optimized regularization parameters (L1 and L2 penalty terms) via nested ten-fold cross-validation (*Cawley and Talbot, 2010*; *Parvandeh et al., 2020*; *Tenenbaum et al., 2000*; *Varma and Simon, 2006*; *Zou and Hastie, 2005*). We split the dataset into training (9/10) and test (1/10) partitions, and each training partition was further split into inner training and testing folds using another ten-fold cross-validation. Within the inner fold, elastic net regularization finds a set of non-redundant features to explain the dependent variable. Using a linear regression, we predicted the IQ scores of

inner fold test data using the features of the selected brain regions by controlling for age, sex, site, and head motion. The model with minimum MAE across the inner folds was applied to the test partition of the outer fold, and the IQ scores of outer fold test data were predicted. The prediction procedure was repeated 100 times with different training and test sets to reduce subject selection bias. Prediction accuracy was indexed by computing linear correlations between the actual and predicted IQ scores as well as MAE. A 95% confidence interval of the accuracy measures was also reported. Permutation-based correlations across 1000 tests were conducted by randomly shuffling subject indices to check whether the prediction performance exceeded chance levels. To assess whether our model outperforms baseline model, we predicted IQ of test data using average of IQ of training data (i.e., predicted IQ = mean(training set IQ)). The improvement of prediction performance was assessed using Meng's z-test (*Meng et al., 1992*). In addition to predicting future IQ, we performed the same prediction analysis to predict the change of IQ between the baseline and follow-up.

## Sensitivity analysis

### Spatial scale

To assess the consistency of our findings across spatial scales, we additionally performed the linear mixed effect modeling using a finer parcellation scheme of 300 parcels (*Figure 1—figure supplement 4*; *Schaefer et al., 2018*).

### Site and sex effect

Participants were recruited from three different sites. To assess whether the longitudinal changes in manifold eccentricity across age are consistent across different sites, we calculated interaction effects of the relationship between age and manifold eccentricity of the identified regions across sites (*Figure 1—figure supplement 5B*). In addition, we computed interaction effect of the relationship between age and manifold eccentricity across male and female subjects to assess whether the age-related changes are affected by biological sexes (*Figure 1—figure supplement 5C*).

### Different parameters for diffusion map embedding

To assess the sensitivity of our findings, we generated connectome manifolds with different parameters for diffusion map embedding ($\alpha$ = 0.25, 0.5, 0.75; t = 0, 1, 2, 3). We assessed age-related changes of the newly defined manifold eccentricity and calculated linear correlation with t-statistic map of the default setting ($\alpha$ = 0.5; t = 0; *Figure 1C*).

### Gradient alignment fidelity

To assess robustness of individual alignment, we computed linear correlations between the template and individual manifolds before and after alignment. We also repeated the linear mixed effect modeling after excluding 10% of subjects with the lowest alignment to the template manifold (*Figure 1—figure supplement 6*).

### Connectome manifold generation using principal component analysis

To explore consistency of our results when using different dimensionality reduction techniques, we generated connectome manifolds using principal component analysis (*Wold et al., 1987*), instead of relying on diffusion map embedding (*Coifman and Lafon, 2006*), and performed longitudinal modeling (*Figure 1—figure supplement 7*). We compared the eigenvectors estimated from diffusion map embedding and principal component analysis using linear correlations.

### Longitudinal changes in graph-theoretical measures

To compare longitudinal changes in manifold eccentricity with those in graph-theoretical centrality measures, we calculated betweenness, degree, and eigenvector centrality of the structural connectomes and built similar linear mixed effects models to assess longitudinal change (*Figure 1—figure supplement 8*). Betweenness centrality is the number of weighted shortest paths between any combinations of nodes that run through that node, degree centrality is the sum of edge weights connected to a given node, and eigenvector centrality measures the influence of a node in the whole network (*Lohmann et al., 2010*; *Rubinov and Sporns, 2010*; *Zuo et al., 2012*). Spatial similarity

between t-statistics of centrality and manifold measures was assessed with 1000 spin tests that account for spatial autocorrelation (*Alexander-Bloch et al., 2018*).

### Manifold eccentricity analysis based on all eigenvectors
We repeated our analysis by calculating manifold eccentricity from all eigenvectors to assess consistency of the findings (*Figure 1—figure supplement 9*).

### Robustness of group representative structural connectome
We compared the distance-dependent thresholding (*Betzel et al., 2019*) that was adopted for the main analysis with a consistency thresholding approach (*Wang et al., 2019*). The latter averages subject-specific matrices, in addition to performing a 50, 40, 30, 20, and 10% thresholding, as well as simple averaging (i.e., 0% thresholding) (*Figure 1—figure supplement 10*).

### Connectome manifolds based on structural parcellation
To confirm whether functional and structural parcellation schemes yield consistent results, we repeated our main analyses using 200 cortical nodes structural parcellation scheme, which preserves the macroscopic boundaries of the Desikan–Killiany atlas (*Desikan et al., 2006*; *Vos de Wael et al., 2020a*; *Figure 1—figure supplement 11*).

### Longitudinal modeling using edge weights
In addition to the analyses based on manifold eccentricity, linear mixed effect modeling using connectome edge weights assessed age-related longitudinal changes in streamline strength (*Figure 1—figure supplement 12*).

### Manifold eccentricity and pubertal stages
To assess the relationship between manifold eccentricity and pubertal stages, we selected a subset of participants who completed Tanner scale (*Marshall and Tanner, 1970*; *Marshall and Tanner, 1969*), which quantifies pubertal stages from 1 (pre-puberty) to 5 (final phase of physical maturation). However, the score was collected at baseline and for 73/208 participants only. To confirm robustness, we performed linear mixed effect modeling using this subset (*Figure 1—figure supplement 13A*). In addition, we assessed interaction effects of Tanner scale and manifold eccentricity restricted to the regions identified from the overall sample (*Figure 1—figure supplement 13B*).

### IQ prediction using nonlinear model
We additionally predicted future IQ score using decision tree learning, a nonlinear approach that builds a regression tree model a root node and split leaf nodes, where the leaf nodes contain the response variables (*Breiman et al., 1984*; *Figure 5—figure supplement 2*).

## Data and code availability
The imaging and phenotypic data were provided by the NSPN 2400 cohort. As stated in https://doi.org/10.1093/ije/dyx117, the NSPN project is committed to make the anonymised dataset fully available to the research community, and participants have consented to their de-identified data being made available to other researchers. A data request can be made to openNSPN@medschl.cam.ac.uk. Codes for connectome manifold generation are available at https://doi.org/10.1038/s42003-020-0794-7; https://github.com/MICA-MNI/BrainSpace (copy archived at swh:1:rev:1fb001f4961d3c0b05b7715f42bcc362b31b96a5; *Vos de Wael et al., 2020b*), and those for calculating manifold eccentricity and subcortical-weighted manifold, as well as performing linear mixed effect modeling to assess age-effects on these features, at out GitHub (https://github.com/MICA-MNI/micaopen/tree/master/manifold_features; copy archived at swh:1:rev:d3988d51e01940007595761dab6b846ce2506433; *Park, 2021*). Source data are provided with this paper.

## Acknowledgements

Dr. Bo-yong Park was funded by the National Research Foundation of Korea (NRF-2020R1A6A3A03037088), Molson Neuro-Engineering fellowship by Montreal Neurological Institute and Hospital (MNI), and a postdoctoral fellowship of the Fonds de la Recherche du Quebec – Santé (FRQ-S). Dr. Richard AI Bethlehem was funded by a British Academy Post-Doctoral Fellowship and the Autism Research Trust. Dr. Casey Paquola and Dr Raul R Cruces were funded through postdoctoral fellowships of the FRQ-S. Ms. Sara Larivière acknowledges funding from the Canadian Institutes of Health Research (CIHR). Dr. Edward T Bullmore was supported by a Senior Investigator award from the National Institute of Health Research (NIHR). Dr. Boris C Bernhardt acknowledges research support from the National Science and Engineering Research Council of Canada (NSERC Discovery-1304413), the CIHR (FDN-154298), SickKids Foundation (NI17-039), Azrieli Center for Autism Research (ACAR-TACC), BrainCanada, FRQ-S, and the Tier-2 Canada Research Chairs program. Drs. Bo-yong Park, Richard A I Bethlehem, Casey Paquola, and Boris C Bernhardt are jointly funded through an MNI-Cambridge collaborative award. The Neuroscience and Psychiatry Network (NSPN) study was funded by a Wellcome Trust award to the University of Cambridge and University College London. The data were curated and analyzed using a computational facility funded by an MRC research infrastructure award (MR/M009041/1) and supported by the NIHR Cambridge Biomedical Research Centre. The views expressed are those of the authors and not necessarily those of the NHS, the NIHR, or the Department of Health and Social Care.

## Additional information

### Group author details

**Neuroscience in Psychiatry Network (NSPN) Consortium**
Edward Bullmore: Department of Psychiatry, University of Cambridge, Cambridge, United Kingdom; Behavioural and Clinical Neuroscience Institute, University of Cambridge, Cambridge, United Kingdom; ImmunoPsychiatry, GlaxoSmithKline Research and Development, GlaxoSmithKline Research Development, , United Kingdom; Raymond Dolan: Max Planck University College London Centre for Computational Psychiatry and Ageing Research, University College London, London, United Kingdom; Wellcome Centre for Human Neuroimaging, University College London, London, United Kingdom; Ian Goodyer: Department of Psychiatry, University of Cambridge, Cambridge, United Kingdom; Peter Fonagy: Research Department of Clinical, Educational and Health Psychology, University College London, London, United Kingdom; Peter Jones: Department of Psychiatry, University of Cambridge, Cambridge, United Kingdom; Michael Moutoussis: Max Planck University College London Centre for Computational Psychiatry and Ageing Research, University College London, London, United Kingdom; Wellcome Centre for Human Neuroimaging, University College London, London, United Kingdom; Tobias Hauser: Max Planck University College London Centre for Computational Psychiatry and Ageing Research, University College London, London, United Kingdom; Wellcome Centre for Human Neuroimaging, University College London, London, United Kingdom; Sharon Neufeld: Department of Psychiatry, University of Cambridge, Cambridge, United Kingdom; Rafael Romero-Garcia: Department of Psychiatry, University of Cambridge, Cambridge, United Kingdom; Behavioural and Clinical Neuroscience Institute, University of Cambridge, Cambridge, United Kingdom; Michelle St Clair: Department of Psychiatry, University of Cambridge, Cambridge, United Kingdom; Petra Vértes: Department of Psychiatry, University of Cambridge, Cambridge, United Kingdom; Behavioural and Clinical Neuroscience Institute, University of Cambridge, Cambridge, United Kingdom; Kirstie Whitaker: Department of Psychiatry, University of Cambridge, Cambridge, United Kingdom; Behavioural and Clinical Neuroscience Institute, University of Cambridge, Cambridge, United Kingdom; Becky Inkster: Department of Psychiatry, University of Cambridge, Cambridge, United Kingdom; Gita Prabhu: Max Planck University College London Centre for Computational Psychiatry and Ageing Research, University College London, London, United Kingdom; Wellcome Centre for Human Neuroimaging, University College London, London, United Kingdom; Cinly Ooi: Department of Psychiatry, University of Cambridge, Cambridge, United Kingdom; Umar Toseeb: Department of

Psychiatry, University of Cambridge, Cambridge, United Kingdom; Barry Widmer: Department of Psychiatry, University of Cambridge, Cambridge, United Kingdom; Junaid Bhatti: Department of Psychiatry, University of Cambridge, Cambridge, United Kingdom; Laura Villis: Department of Psychiatry, University of Cambridge, Cambridge, United Kingdom; Ayesha Alrumaithi: Department of Psychiatry, University of Cambridge, Cambridge, United Kingdom; Sarah Birt: Department of Psychiatry, University of Cambridge, Cambridge, United Kingdom; Aislinn Bowler: Wellcome Centre for Human Neuroimaging, University College London, London, United Kingdom; Kalia Cleridou: Wellcome Centre for Human Neuroimaging, University College London, London, United Kingdom; Hina Dadabhoy: Wellcome Centre for Human Neuroimaging, University College London, London, United Kingdom; Emma Davies: Department of Psychiatry, University of Cambridge, Cambridge, United Kingdom; Ashlyn Firkins: Department of Psychiatry, University of Cambridge, Cambridge, United Kingdom; Sian Granville: Wellcome Centre for Human Neuroimaging, University College London, London, United Kingdom; Elizabeth Harding: Wellcome Centre for Human Neuroimaging, University College London, London, United Kingdom; Alexandra Hopkins: Max Planck University College London Centre for Computational Psychiatry and Ageing Research, University College London, London, United Kingdom; Wellcome Centre for Human Neuroimaging, University College London, London, United Kingdom; Daniel Isaacs: Wellcome Centre for Human Neuroimaging, University College London, London, United Kingdom; Janchai King: Wellcome Centre for Human Neuroimaging, University College London, London, United Kingdom; Danae Kokorikou: Wellcome Centre for Human Neuroimaging, University College London, London, United Kingdom; Research Department of Clinical, Educational and Health Psychology, University College London, London, United Kingdom; Christina Maurice: Department of Psychiatry, University of Cambridge, Cambridge, United Kingdom; Cleo McIntosh: Department of Psychiatry, University of Cambridge, Cambridge, United Kingdom; Jessica Memarzia: Department of Psychiatry, University of Cambridge, Cambridge, United Kingdom; Harriet Mills: Wellcome Centre for Human Neuroimaging, University College London, London, United Kingdom; Ciara O'Donnell: Department of Psychiatry, University of Cambridge, Cambridge, United Kingdom; Sara Pantaleone: Wellcome Centre for Human Neuroimaging, University College London, London, United Kingdom; Jenny Scott: Department of Psychiatry, University of Cambridge, Cambridge, United Kingdom; Beatrice Kiddle: Department of Psychiatry, University of Cambridge, Cambridge, United Kingdom; Ela Polek: Department of Psychiatry, University of Cambridge, Cambridge, United Kingdom; Pasco Fearon: Research Department of Clinical, Educational and Health Psychology, University College London, London, United Kingdom; John Suckling: Department of Psychiatry, University of Cambridge, Cambridge, United Kingdom; Anne-Laura van Harmelen: Department of Psychiatry, University of Cambridge, Cambridge, United Kingdom; Rogier Kievit: Max Planck University College London Centre for Computational Psychiatry and Ageing Research, University College London, London, United Kingdom; Medical Research Council Cognition and Brain Sciences Unit, University of Cambridge, Cambridge, United Kingdom; Sam Chamberlain: Department of Psychiatry, University of Cambridge, Cambridge, United Kingdom

## Competing interests

Edward T Bullmore: ETB. serves on the scientific advisory board of Sosei Heptares and as a consultant for GlaxoSmithKline. The other authors declare that no competing interests exist.

## Funding

| Funder | Grant reference number | Author |
| --- | --- | --- |
| Canada Research Chairs | | Boris C Bernhardt |
| National Research Foundation of Korea | NRF2020R1A6A3A03037088 | Bo-yong Park |
| Fonds de la Recherche du Quebec – Santé | | Bo-yong Park<br>Casey Paquola<br>Raul Rodríguez-Cruces<br>Boris C Bernhardt |

| Montreal Neurological Institute and Hospital (MNI) | Molson Neuro-Engineering fellowship | Bo-yong Park |
| --- | --- | --- |
| British Academy | Post-Doctoral Fellowship | Richard AI Bethlehem |
| Autism Research Trust | | Richard AI Bethlehem |
| Canadian Institutes of Health Research | | Sara Larivière |
| National Institute for Health Research | Senior Investigator award | Edward T Bullmore |
| Natural Sciences and Engineering Research Council of Canada | NSERC Discovery-1304413 | Boris C Bernhardt |
| Canadian Institutes of Health Research | FDN-154298 | Boris C Bernhardt |
| SickKids Foundation | NI17-039 | Boris C Bernhardt |
| Azrieli Center for Autism Research | ACAR-TACC | Boris C Bernhardt |
| BrainCanada | | Boris C Bernhardt |
| MNI-Cambridge collaborative award | | Bo-yong Park Richard AI Bethlehem Casey Paquola Boris C Bernhardt |

The funders had no role in study design, data collection and interpretation, or the decision to submit the work for publication.

## Author contributions

Bo-yong Park, Conceptualization, Software, Investigation, Visualization, Methodology, Writing - original draft, Writing - review and editing; Richard AI Bethlehem, Casey Paquola, Conceptualization, Methodology, Writing - review and editing; Sara Larivière, Raul Rodríguez-Cruces, Reinder Vos de Wael, Methodology, Writing - review and editing; Neuroscience in Psychiatry Network (NSPN) Consortium, Data curation; Edward T Bullmore, Conceptualization, Resources, Methodology, Writing - review and editing; Boris C Bernhardt, Conceptualization, Supervision, Writing - original draft, Writing - review and editing

## Author ORCIDs

Bo-yong Park  https://orcid.org/0000-0001-7096-337X
Richard AI Bethlehem  https://orcid.org/0000-0002-0714-0685
Casey Paquola  http://orcid.org/0000-0002-0190-4103
Boris C Bernhardt  https://orcid.org/0000-0001-9256-6041

## Ethics

Human subjects: Participants provided informed written consent for each aspect of the study, and parental consent was obtained for those aged 14-15 years old. Ethical approval was granted for this study by the NHS NRES Committee East of England-Cambridge Central (project ID 97546). The authors assert that all procedures contributing to this work comply with the ethical standards of the relevant national and institutional committees on human experimentation and with the Helsinki Declaration of 1975, as revised in 2008.

## Decision letter and Author response

Decision letter https://doi.org/10.7554/eLife.64694.sa1
Author response https://doi.org/10.7554/eLife.64694.sa2

## Additional files

### Supplementary files

• Supplementary file 1. Significant gene lists correlated with patterns of manifold eccentricity changes across age. Gene symbols with name and t-statistic as well as false discovery rate-corrected p-value are reported in the Supplementary File (Supplementary_File1.xlsx).

• Supplementary file 2. Full details for Neuroscience in Psychiatry Network (NSPN) Consortium.

• Transparent reporting form

### Data availability

The imaging and phenotypic data were provided by the NSPN 2400 cohort. As stated in https://doi. org/10.1093/ije/dyx117, the NSPN project is committed to make the anonymised dataset fully available to the research community, and participants have consented to their de-identified data being made available to other researchers. A data request can be made to openNSPN@medschl.cam.ac. uk. Codes for connectome manifold generation are available at https://doi.org/10.1038/s42003-020-0794-7; https://github.com/MICA-MNI/BrainSpace (copy archived at https://archive.softwareheritage.org/swh:1:rev:1fb001f4961d3c0b05b7715f42bcc362b31b96a5/), and those for calculating manifold eccentricity and subcortical-weighted manifold, as well as performing linear mixed effect modeling to assess age-effects on these features at our GitHub (https://github.com/MICA-MNI/micaopen/tree/master/manifold_features; copy archived at https://archive.softwareheritage.org/swh:1:rev:d3988d51e01940007595761dab6b846ce2506433/).

The following datasets were generated:

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
