## [Decision Letter]

**Acceptance summary:**

This manuscript describes a longitudinal study of the adolescent structural connectome. The authors find strong effects of expansion of structural connectomes in transmodal brain regions during adolescence. They also report findings centered on the caudate and thalamus, and supplement the structural connectivity analyses with transcriptome association analyses revealing genes enriched in specific brain regions. Finally, intelligence measures are predicted from baseline structural measures. These findings help to bridge large-scale brain network configurations, microscale processes, and cognition in adolescent development.

**Decision letter after peer review:**

Thank you for submitting your article "An expanding manifold characterizes adolescent reconfigurations of structural connectome organization" for consideration by *eLife*. Your article has been reviewed by 3 peer reviewers, and the evaluation has been overseen by a Reviewing Editor and Timothy Behrens as the Senior Editor. The following individuals involved in review of your submission have agreed to reveal their identity: Ben Fulcher (Reviewer #3).

The reviewers have discussed the reviews with one another and the Reviewing Editor has drafted this decision to help you prepare a revised submission.

Summary:

This manuscript describes a longitudinal study of the adolescent structural connectome. Park et al. report on an analysis of existing semi-longitudinal NSPN 2400 data to learn how the projections of high-dimensional structural connectivity patterns onto a three-dimensional subspace change with age during adolescence. They employ a non-linear manifold learning algorithm (diffusion embedding), thereby linking the maturation of global structural connectivity patterns to an emerging approach in understanding brain organization through spatial gradient representations. The authors find strong effects of expansion of structural connectomes in transmodal brain regions during adolescence. They also report findings centered on the caudate and thalamus, and supplement the structural connectivity analyses with transcriptome association analyses revealing gens enriched in specific brain regions. Finally, intelligence measures are predicted from baseline structural measures.

This is an interesting and comprehensive set of analyses on an important topic. Overall, the figures are lovely. The sensitivity analyses are particularly commendable.

The paper is well written, the data are fantastic, and the analyses are interesting. Some suggestions and points for clarification (both theoretical and methodological) are below.

Essential revisions:

– There is not much in the introduction about why co-localized gene sets are of interest to explore. What is already known about brain development using this approach, and how does the current work fill a gap in our knowledge?

– Similarly, the introduction states that the study aims to "predict future measures of cognitive function". What cognitive functions specifically were of interest in this study, and why? No rationale or background is provided for conducting these analyses.

– The authors claim that their study examines "the entire adolescent time period", however some would argue that age 14 does not represent the earliest age at which adolescence onsets. I think it would be more accurate to say the study covers the mid to late adolescent period.

– In the results it is stated that three eigenvector explained approximately 50% of the variance in the template affinity matrix. Here it would be helpful to report exactly how much of the variance was explained by each (E1, E2, E3).

– Pubertal development occurs across the age range investigated, and affects brain structure and function. Was information on pubertal stage of participants available? Did some participants undergo changes in pubertal status from timepoint 1 to timepoint 2?

– The introduction does not mention cortical thickness much, therefore these analyses come as a bit of surprise in the results.

– As in the introduction, there is not much interpretation of the transcriptome findings in the discussion.

– For constructing the structural connectome, the Schaefer 7-network atlas was utilized. Can the authors comment on why a functional atlas (rather than a structural atlas) was used here?

– While this work touches on an important topic, ties nicely with the increasing body of papers on global brain gradients, and its overall conclusions are warranted, I am not (yet) convinced that it offers fundamentally new insights that could not have been gleaned from previous work (after all, manifold learning simply displays a shadow of the underlying patterns; if the patterns change, so does their shadow). I am also not convinced by the rationale for employing diffusion embedding: the authors state that the ensuing gradients are heritable, conserved across species, capture functional activation patterns during task states, and provide a coordinate system to interrogate brain structure and function, but that would be true for any method that adequately captures biologically meaningful variance in the structural connectivity patterns.

– The authors show that the maturational change of the manifold features predict intelligence at follow-up but did not show that intelligence itself exhibited changes that exceeded the error bounds of the regression line. Why not predict IQ change?

– The slight improvements in prediction accuracy observed after adding maturational change and subcortical features to the features at baseline will necessarily happen by adding more regression parameters and may not be meaningful.

– Interpretability and Lack of Comparison

The authors claim repeatedly that they are "capitalizing on advanced manifold learning techniques".

One could imagine an infinite number of papers that take a dataset, use a technique to extract a metric, X (e.g., eccentricity), and then write about the changes in X with some property of interest, Y (e.g., age). Given this set of papers (and the non-independence between the set of possible Xs), the reader ought to be most interested in those Xs that provide the best performance and simplest interpretation, with other papers being redundant.

Thus, a nuanced approach to presenting a paper like this is to demonstrate that the metric used represents an advance over alternative, simpler-to-compute, or clearer-to-interpret metrics that already exist.

In this paper, however, the authors do not demonstrate the benefits of their particular choice of applying a specific nonlinear dimensionality reduction method using 3 dimensions alignment to a template manifold and then computing an eccentricity metric. For example:

i. Is the nonlinearity required (e.g., does it outperform PCA or MDS)?

ii. Is there something special about picking 3 dimensions to do the eccentricity calculation? Is dimensionality reduction required at all (e.g., would you get similar results by computing eccentricity in the full-dimensional space?)

iii. Does it outperform basic connectome measures (e.g., the simple ones the authors compute)?

There is a clear down-side of how opaque the approach is (and thus difficult to interpret relative to, say, connectivity degree), so one would hope for a correspondingly strong boost in performance. The authors could also do more to develop some intuition for the idea of a low-dimensional connection-pattern-similarity-space, and how to interpret taking Euclidean distances within such a space.

– Developmental Enrichment Analysis

Both in the main text and in the Materials and methods, this is described as "genes were fed into a developmental enrichment analysis". Can some explanation be provided as to what happens between the "feeding in" and what comes out? Without clearly described methods, it is impossible to interpret or critique this component of the paper. If the methodological details are opaque, then the significance of the results could be tested numerically relative to some randomized null inputs being 'fed in' to demonstrate specificity of the tested phenotype.

– IQ prediction

The predictions seem to be very poor (equality lines, y = x, should be drawn in Figure 5, to show what perfect predictions would look like; linear regressions are not helpful for a prediction task, and are deceptive of the appropriate MAE computation). The authors do not perform any comparisons in this section (even to a real baseline model like `predicted_IQ = mean(training_set_IQ)`). They also do not perform statistical tests (or quote p-values), but nevertheless make a range of claims, including of "significant prediction" or "prediction accuracy was improved", "reemphasize the benefits of incorporating subcortical nodes", etc. All of these claims should be tested relative to rigorous statistics, and comparisons to appropriate baseline/benchmark approaches.

– Group Connectome

Given how much the paper relies on estimating a group structural connectome, it should be visualized and characterized. For example, a basic analysis of the distribution of edge weights and degree, especially as edge weights can vary over orders of magnitude and high weights (more likely to be short distances) may therefore unduly dominate some of the low-dimensional components. The authors may also consider testing robustness performed to alternative ways of estimating the connectome [e.g., Oldham et al. NeuroImage 222, 117252 (2020)] and its group-level summary [e.g., Roberts et al. NeuroImage 145, 1-42 (2016)].

– Individual Alignment

The paper relies on individuals being successfully aligned to the template manifold. Accordingly, some analysis should be performed quantifying how well individuals could be mapped. Presumably some subjects fit very well onto the template, whereas others do not. Is there something interesting about the poorly aligned subjects? Do your results improve when excluding them?

---

## [Author Response]

Essential revisions:– There is not much in the introduction about why co-localized gene sets are of interest to explore. What is already known about brain development using this approach, and how does the current work fill a gap in our knowledge?

We updated Introduction by adding a new paragraph motivating the transcriptomic analysis and outlining the novelty of our work:

“Imaging-transcriptomics approaches allow for the identification of cellular and molecular factors that co-vary with imaging-based findings (Arnatkeviciute et al., 2019; Fornito et al., 2019; Gorgolewski et al., 2014; Hawrylycz et al., 2015; Thompson et al., 2013). Recently established resources, such as the Allen Human Brain Atlas (Arnatkeviciute et al., 2019; Hawrylycz et al., 2015), can be utilized to spatially associate macroscale imaging/connectome data with the expression patterns of thousands of genes. These findings have already been applied in the study of healthy adults (Hawrylycz et al., 2015; Park et al., 2020b; Q. Xu et al., 2020) and typically developing adolescents (Mascarell Maričić et al., 2020; Padmanabhan and Luna, 2014; Paquola et al., 2019a; Vértes et al., 2016; Whitaker et al., 2016), as well as individuals suffering from prevalent brain disorders (Altmann et al., 2018; Hashimoto et al., 2015; Klein et al., 2017; Park et al., 2020a; Patel et al., 2021; Romero-Garcia et al., 2019). The gene sets that co-vary with in vivo findings can furthermore be subjected to gene set enrichment analyses to discover potentially implicated molecular, cellular, and pathological processes (Ashburner et al., 2000; Carbon et al., 2019; Chen et al., 2013; Dougherty et al., 2010; Kuleshov et al., 2016; Morgan et al., 2019; Romero-Garcia et al., 2018; Subramanian et al., 2005). For example, studies in newborns have shown that cortical morphology reflects spatiotemporal patterns of gene expression in foetuses, linking molecular mechanisms to in vivo measures of cortical development in early life (Ball et al., 2020). Work in adolescents has furthermore shown that developmental changes in regional cortical thickness measures and myelin proxies spatially co-localize with the expression patterns of genes involved in synaptic and oligodendroglial function (Paquola et al., 2019a; Whitaker et al., 2016). Building on these prior investigations, the current study aimed at exploring whether adolescent structural connectome reconfigurations, assessed using manifold learning techniques, reflect the expression patterns of specific genes, in order to identify potential molecular signatures of macroscale structural network development.”

– Similarly, the introduction states that the study aims to "predict future measures of cognitive function". What cognitive functions specifically were of interest in this study, and why? No rationale or background is provided for conducting these analyses.

We thank the Reviewers for this comment.

We incorporated additional justification and details on the cognitive prediction in the Introduction:

“To also assess behavioral associations of connectome manifold changes, we utilized supervised machine learning to predict future measures of cognitive function quantified via the intelligence quotient (IQ). IQ is a widely used marker of general cognitive abilities, which shows good test-retest reliability (Brown and May, 1979; Canivez and Watkins, 1998; Catron, 1978; Matarazzo et al., 1973; Snow et al., 1989; Wagner and Caldwell, 1979) and has previously been applied to index overall cognitive function during development (Crespi, 2016; Garde et al., 2005, 2000; Koenis et al., 2018; Park et al., 2016; Ramsden et al., 2011; Shaw et al., 2006; Suprano et al., 2020). In the study of neurodevelopment, neuroimaging reports have previously assessed associations between IQ and large-scale network measures in children to adolescents (Koenis et al., 2018; Ramsden et al., 2011; Seidlitz et al., 2018; Shaw et al., 2006; Suprano et al., 2020).”

– The authors claim that their study examines "the entire adolescent time period", however some would argue that age 14 does not represent the earliest age at which adolescence onsets. I think it would be more accurate to say the study covers the mid to late adolescent period.

We appreciate the Reviewer’s comment.

We clarified the sample in the Introduction:

“Here, we charted developmental changes in structural connectome organization, based on an accelerated longitudinal neuroimaging study involving 208 participants investigated between 14 to 26 years of age (Kiddle et al., 2018; Whitaker et al., 2016)”

– In the results it is stated that three eigenvector explained approximately 50% of the variance in the template affinity matrix. Here it would be helpful to report exactly how much of the variance was explained by each (E1, E2, E3).

As suggested, we provided further details in the Results:

“Three eigenvectors (E1, E2, and E3) explained approximately 50% of information in the template affinity matrix (i.e., 20.7/15.8/13.5% for E1/E2/E3, respectively), with each eigenvector showing a different axis of spatial variation across the cortical mantle (Figure 1A).”

– Pubertal development occurs across the age range investigated, and affects brain structure and function. Was information on pubertal stage of participants available? Did some participants undergo changes in pubertal status from timepoint 1 to timepoint 2?

The NSPN consortium collected Tanner scale (Marshall and Tanner, 1970, 1969), which quantifies pubertal stages from 1 (pre-puberty) to 5 (final phase of physical maturation). However, the score was collected at baseline and for 73/208 participants only. We could thus not assess longitudinal changes in manifold features according to changes in pubertal stages indexed by Tanner staging. Nevertheless, linear mixed effect modeling in the subset of these 73 participants confirmed our overall findings. While the statistical power was reduced in this smaller subset, spatial patterns of age-effects were consistent with the larger sample (Figure 1—figure supplement 13A). Notably, manifold eccentricity within the identified regions derived from overall sample and Tanner scale revealed significant interaction effect (t = 2.36, p = 0.01; Figure 1—figure supplement 13B), suggesting participants in early pubertal stage show stronger changes in manifold eccentricity across age compared to those in later stages.

We updated the Results:

“k) Manifold eccentricity and pubertal stages. We repeated the longitudinal modeling within a subset of participants who completed the Tanner scale (n = 73) (Marshall and Tanner, 1970, 1969), and found relatively consistent albeit weaker age-related changes in manifold eccentricity as for the overall sample (Figure 1—figure supplement 13A). Notably, manifold eccentricity within the identified regions derived from overall sample and Tanner scale revealed a significant interaction effect (t = 2.36, p = 0.01; Figure 1—figure supplement 13B), suggesting participants in early pubertal stages show more marked changes in manifold eccentricity across age compared to those in later stages.”

as well as Materials and methods:

“k) Manifold eccentricity and pubertal stages. To assess the relationship between manifold eccentricity and pubertal stages, we selected a subset of participants who completed Tanner scale (Marshall and Tanner, 1970, 1969), which quantifies pubertal stages from 1 (pre-puberty) to 5 (final phase of physical maturation). However, the score was collected at baseline and for 73/208 participants only. To confirm robustness, we performed linear mixed effect modeling using this subset (Figure 1—figure supplement 13A). In addition, we assessed interaction effects of Tanner scale and manifold eccentricity restricted to the regions identified from the overall sample (Figure 1—figure supplement 13B).”

– The introduction does not mention cortical thickness much, therefore these analyses come as a bit of surprise in the results.

As suggested, we now contextualize adolescent cortical thickness changes in the Introduction:

“Early cross-sectional and longitudinal studies in neurodevelopmental cohorts focused on the analysis of morphological changes (Gogtay et al., 2004; Shaw et al., 2006; Tamnes et al., 2017), including MRI-based cortical thickness (Shaw et al., 2006; Tamnes et al., 2017) and volumetric measures (Gogtay et al., 2004; Tamnes et al., 2017). Studies robustly show initial grey matter increases until mid-late childhood followed by a decline for the rest of the lifespan. During adolescence, cortical thickness decreases in widespread brain regions (Khundrakpam et al., 2013; Shaw et al., 2006; Sotiras et al., 2017; Tamnes et al., 2017). Thus, contextualizing connectome alterations relative to established patterns of cortical thickness findings may establish whether inter-regional network changes occur above and beyond these diffuse effects of regional morphological maturation.”

– As in the introduction, there is not much interpretation of the transcriptome findings in the discussion.

As suggested, we expanded our interpretation of the transcriptome findings in the Discussion:

“Associating macroscopic changes in manifold eccentricity with post-mortem microarray data provided by the Allen Human Brain Atlas (Arnatkeviciute et al., 2019; Fornito et al., 2019; Gorgolewski et al., 2014; Hawrylycz et al., 2015; Thompson et al., 2013), we identified gene sets expressed in cortical regions and subcortical structures of the thalamus and striatum during late childhood, adolescence, and young adulthood. Despite these findings being associative and based on separate datasets, they overall support our results that brain network maturation from late childhood until early adulthood implicates micro- and macroscale factors in both subcortical and cortical networks. Coupled network and molecular changes may ultimately change subcortical and cortical circuit properties, including the balance of excitation and inhibition (E/I). Human brain development involves spatio-temporal waves of gene expression changes across different brain regions and developmental time windows (Ip et al., 2010; Kang et al., 2011; Shin et al., 2018). In the study of adolescent development, prior studies have suggested shifts in E/I balance, evolving from a dominant inhibitory bias in early developmental stages towards stronger excitatory drive in later stages, and suggested that these may underlie the maturation of cognitive functions such as working memory and executive control (Dorrn et al., 2010; Lander et al., 2017; Liu et al., 2007). In common neurodevelopmental disorders, including autism, schizophrenia, and attention deficit hyperactivity disorder, imbalances in cortical E/I and cortico-subcortical network function have been demonstrated (Cellot and Cherubini, 2014; Gandal et al., 2018; Lee et al., 2017; Lewis et al., 2005; Nelson and Valakh, 2015; Park et al., 2020a; Sohal and Rubenstein, 2019; Trakoshis et al., 2020), potentially downstream to perturbations of different neurotransmitter systems, such as interneuron-mediated GABA transmission (Bonaventura et al., 2017; Kilb, 2012; Liu et al., 2007; Park et al., 2020a; Silveri et al., 2013; Trakoshis et al., 2020; Tziortzi et al., 2014).”

– For constructing the structural connectome, the Schaefer 7-network atlas was utilized. Can the authors comment on why a functional atlas (rather than a structural atlas) was used here?

We opted for the widely used Schaefer parcellation (Schaefer et al., 2018) as it (i) allows contextualization of our findings within the context of intrinsic functional communities (Yeo et al., 2011), (ii) incorporates the option to assess results across different parcel numbers, and (iii) aligns the current study with prior work from our group (Benkarim et al., 2020; Paquola et al., 2020; Park et al., 2021, 2020a; Rodríguez-Cruces et al., 2020) and others (Baum et al., 2020; Betzel et al., 2019; Osmanlıoğlu et al., 2019).

This justification is now included in the Materials and methods:

“We opted for this atlas as it (i) allows contextualization of our findings within macroscale intrinsic functional communities (Yeo et al., 2011), (ii) incorporates the option to assess results across different granularities, and (iii) aligns the current study with previous work from our group (Benkarim et al., 2020; Paquola et al., 2020; Park et al., 2021, 2020a; Rodríguez-Cruces et al., 2020) and others (Baum et al., 2020; Betzel et al., 2019; Osmanlıoğlu et al., 2019).”

We agree with the reviewer that diffusion MRI studies have often been conducted based on structural atlases. We thus repeated our analyses using a structural parcellation scheme with 200 nodes that preserves the macroscopic boundaries of the Desikan-Killiany atlas (Desikan et al., 2006; Vos de Wael et al., 2020). We found largely consistent age-effects on manifold eccentricity and subcortical-weighted manifold in heteromodal association areas, as well as in caudate, hippocampus, and thalamus (Figure 1—figure supplement 11).

We updated Results:

“i) Connectome manifolds based on structural parcellation. We repeated our analyses with a structural parcellation, defined using a sub-parcellation of folding based on the Desikan-Killiany atlas (Desikan et al., 2006; Vos de Wael et al., 2020) (Figure 1—figure supplement 11). Despite slight differences in the topography of manifold eccentricity in lateral prefrontal, temporal, and occipital cortices, we could replicate strong age-related effects in heteromodal association areas, together with effects in caudate and hippocampus (FDR < 0.05), and marginally in thalamus (FDR < 0.1).”

as well as Materials and methods:

“i) Connectome manifolds based on structural parcellation. To confirm whether functional and structural parcellation schemes yield consistent results, we repeated our main analyses using 200 cortical nodes structural parcellation scheme, which preserves the macroscopic boundaries of the Desikan-Killiany atlas (Desikan et al., 2006; Vos de Wael et al., 2020) (Figure 1—figure supplement 11).”

– While this work touches on an important topic, ties nicely with the increasing body of papers on global brain gradients, and its overall conclusions are warranted, I am not (yet) convinced that it offers fundamentally new insights that could not have been gleaned from previous work (after all, manifold learning simply displays a shadow of the underlying patterns; if the patterns change, so does their shadow). I am also not convinced by the rationale for employing diffusion embedding: the authors state that the ensuing gradients are heritable, conserved across species, capture functional activation patterns during task states, and provide a coordinate system to interrogate brain structure and function, but that would be true for any method that adequately captures biologically meaningful variance in the structural connectivity patterns.

We thank the reviewer for the positive evaluation of our work and happy to provide further rationale for the use of gradients. Being derived from connectomes, we agree that gradient mapping techniques do, in part, recapitulate findings that have been previously shown with more established techniques, for example, graph-theoretical analyses that consider the brain as a graph with nodes and edges and estimate connectome patterns by calculating graph parameters, such as centrality and modular measures (Bullmore and Sporns, 2009; Rubinov and Sporns, 2010). However, we believe that gradient mapping approaches offer a promising combination of methodological and conceptual properties: (i) Gradient mapping based on manifold learning approaches reduce high dimensional (i.e., *n x n*) connectomes into a series of *k* spatial maps (i.e., gradients, each of the form *n x 1*) in a data-driven way and thus provide a simplified view on large-scale connectome organization. Of note, the dimensionality reduction procedure we opted for (i.e., diffusion map embedding) is computationally efficient and robust to noise when computing a globally optimal solution compared to linear approaches (Tenenbaum et al., 2000). (ii) As in the current work, the solution of these techniques is not necessarily a single gradient, but it can be multiple, potentially overlapping gradients. Representing multiple gradients has been suggested to offer the ability to characterize both subregional heterogeneity and functional multiplicity of brain regions (Haak and Beckmann, 2020). In prior work, we showed that the use of multiple diffusion MRI gradients can help to better understand dynamic functional communication patterns in the adult human brain connectome (Park et al., 2021). (iii) As the gradients derived from these techniques can serve as continuous axes of cortical organization, the use of several gradients jointly allows to generate intrinsic coordinate systems in connectivity space, in line with prior conceptual accounts (Bijsterbosch et al., 2020; Haak et al., 2018; Huntenburg et al., 2018; Margulies et al., 2016; Mars et al., 2018). (iv) In addition to the aforementioned methodological advances, prior work has shown that principal gradients, e.g., those estimated from task-free functional MRI (Margulies et al., 2016), follow established models of cortical hierarchy and laminar differentiation (Mesulam, 1998). These observations have allowed gradient mapping approaches to make conceptual contact with seminal work on cortical organization, evolution, and development (Buckner and Krienen, 2013; Goulas et al., 2018; Huntenburg et al., 2018; Sanides, 1969, 1962). What’s more, an emerging literature has shown utility of the gradient framework to study primate evolution and cross-species alignment (Blazquez Freches et al., 2020; Sofie L. Valk et al., 2020; T. Xu et al., 2020), neurodevelopment (Hong et al., 2019; Paquola et al., 2019a), as well as plasticity and structure-function coupling (Park et al., 2021; Vázquez-Rodríguez et al., 2019).

We updated Introduction:

“One emerging approach to address connectome organization and development comes from the application of manifold learning techniques to connectivity datasets. […] An emerging literature has indeed shown utility of the gradient framework to study primate evolution and cross-species alignment (Blazquez Freches et al., 2020; Sofie L. Valk et al., 2020; T. Xu et al., 2020), neurodevelopment (Hong et al., 2019; Paquola et al., 2019a), as well as plasticity and structure-function coupling (Park et al., 2021; Sofie L Valk et al., 2020; Vázquez-Rodríguez et al., 2019).”

To build cortex-wide structural connectome manifolds, our work opted for diffusion map embedding, a non-linear dimensionality reduction technique (Coifman and Lafon, 2006), following a seminal connectivity gradient study (Margulies et al., 2016). Follow-up work from our group and others adopted the same approach (Hong et al., 2019, 2020; Huntenburg et al., 2017; Larivière et al., 2020; Margulies et al., 2016; Müller et al., 2020; Paquola et al., 2019a; Park et al., 2021; Sofie L. Valk et al., 2020; Vos de Wael et al., 2020). In addition to ensuring continuity with these prior studies, non-linear techniques do in general explain more information with the same number of components than linear techniques, which may increase performance on group classification, cluster identification, and phenotypic score prediction (Errity and McKenna, 2007; Gallos et al., 2020; Hong et al., 2020). Diffusion map embedding, in particular, is only controlled by few parameters, and thus computationally efficient and rather robust to noise (Errity and McKenna, 2007; Gallos et al., 2020; Hong et al., 2020; Tenenbaum et al., 2000). Of note, the non-linearity is simply introduced by running the singular value decomposition, which is also at the heart of PCA, on the Markov matrix of the normalized connectome, which also lends a direct interpretation to the reduced manifold as being interpretable as a ‘diffusion map’ between the nodes.

We nevertheless agree with the Reviewer’s point that non-linear and linear techniques often converge, and that it is not fully established which approach should be prioritized (Hong et al., 2020; Vos de Wael et al., 2020). To assess consistency, we thus repeated our analysis based on principal component analysis (PCA) and observed consistent manifold dimensions (linear correlation = 0.998 ± 0.001 across E1/E2/E3; Figure 1—figure supplement 7A), as well as similar age-effects as in the main analysis, notably a similar expansion of the manifold space (Figure 1—figure supplement 7B).

We updated Results:

“e) Connectome manifold generation using principal component analysis. In a separate analysis, we generated eigenvectors using principal component analysis (Wold et al., 1987), instead of diffusion map embedding (Coifman and Lafon, 2006), and found consistent spatial maps (linear correlation = 0.998 ± 0.001 across E1/E2/E3; Figure 1—figure supplement 7A) and longitudinal findings (Figure 1—figure supplement 7B).”

as well as Materials and methods:

“An affinity matrix was constructed with a normalized angle kernel, and eigenvectors were estimated via diffusion map embedding (Figure 1A), a non-linear dimensionality reduction technique (Coifman and Lafon, 2006) that projects connectome features into low dimensional manifolds (Margulies et al., 2016). This technique is only controlled by few parameters, computationally efficient, and relatively robust to noise compared to other non-linear techniques (Errity and McKenna, 2007; Gallos et al., 2020; Hong et al., 2020; Tenenbaum et al., 2000), and has been extensively used in the previous gradient mapping literature (Hong et al., 2019, 2020; Huntenburg et al., 2017; Larivière et al., 2020a; Margulies et al., 2016; Müller et al., 2020; Paquola et al., 2019a; Park et al., 2021; Sofie L. Valk et al., 2020; Vos de Wael et al., 2020).”

“e) Connectome manifold generation using principal component analysis. To explore consistency of our results when using different dimensionality reduction techniques, we generated connectome manifolds using principal component analysis (Wold et al., 1987), instead of relying on diffusion map embedding (Coifman and Lafon, 2006), and performed longitudinal modeling (Figure 1—figure supplement 7). We compared the eigenvectors estimated from diffusion map embedding and principal component analysis using linear correlations.”

– The authors show that the maturational change of the manifold features predict intelligence at follow-up, but did not show that intelligence itself exhibited changes that exceeded the error bounds of the regression line. Why not predict IQ change?

As suggested, we additionally predicted the change of IQ between the baseline and follow-up, instead of IQ at follow-up, using the imaging features. However, we could not find significant results for predicting ∆IQ using cortical features at baseline (mean ± SD r = -0.10 ± 0.04, MAE = 6.62 ± 0.06, p = 0.27) nor at both baseline and maturational change (r = -0.08 ± 0.04, MAE = 6.60 ± 0.06, p = 0.34). Adding subcortical regions did not improve the performance (baseline: r = -0.01 ± 0.03, MAE = 7.02 ± 0.12, p = 0.73; baseline and maturational change: r = 0.03 ± 0.03, MAE = 6.53 ± 0.06, p = 0.67). The low performance might be due to small variations in ∆IQ, where many participants (42%) showed IQ changes less than 5 and 76% less than 10.

We updated Results:

“We also predicted the change of IQ between the baseline and follow-up, instead of IQ at follow-up, using the imaging features. However, we could not find significant results.”

as well as Materials and methods:

“In addition to predicting future IQ, we performed the same prediction analysis to predict the change of IQ between the baseline and follow-up.”

– The slight improvements in prediction accuracy observed after adding maturational change and subcortical features to the features at baseline will necessarily happen by adding more regression parameters and may not be meaningful.

To improve robustness of prediction analysis, as well as avoid overfitting, the revised work employed a nested ten-fold cross-validation framework (Cawley and Talbot, 2010; Parvandeh et al., 2020; Tenenbaum et al., 2000; Varma and Simon, 2006) with elastic net regularization (Zou and Hastie, 2005). Specifically, we split the dataset into training (9/10) and test (1/10) partitions, and each training partition was further split into inner training and testing folds using another ten-fold cross-validation. Within the inner fold, elastic net regularization finds a set of non-redundant features to explain the dependent variable. Using a linear regression, we predicted the IQ scores of inner fold test data using the features of the selected brain regions by controlling for age, sex, site, and head motion. The model with minimum mean absolute error (MAE) across the inner folds was applied to the test partition of the outer fold, and the IQ scores of outer fold test data were predicted. The prediction procedure was repeated 100 times with different training and test sets to reduce subject selection bias. Across cross-validation and iterations, 6.24 ± 5.74 (mean ± SD) features were selected to predict IQ using manifold eccentricity of cortical regions at baseline, 6.20 ± 5.14 cortical features at baseline and maturational change, 5.45 ± 5.99 cortical and subcortical features at baseline, and 5.16 ± 5.43 at baseline and maturational change. In this scenario, adding more independent variables may not per se lead to improvement in prediction accuracy.

We updated Results:

“We used elastic net regularization with nested ten-fold cross-validation (Cawley and Talbot, 2010; Parvandeh et al., 2020; Tenenbaum et al., 2000; Varma and Simon, 2006; Zou and Hastie, 2005) (see Methods), and repeated the prediction 100 times with different training and test dataset compositions to mitigate subject selection bias. Across cross-validation and iterations, 6.24 ± 5.74 (mean ± SD) features were selected to predict IQ using manifold eccentricity of cortical regions at baseline, 6.20 ± 5.14 cortical features at baseline and maturational change, 5.45 ± 5.99 cortical and subcortical features at baseline, and 5.16 ± 5.43 at baseline and maturational change, suggesting that adding more independent variables may not per se lead to improvement in prediction accuracy.”

as well as Materials and methods:

“For each evaluation, a subset of features that could predict future IQ was identified using elastic net regularization (𝜌 = 0.5) with optimized regularization parameters (L1 and L2 penalty terms) via nested ten-fold cross-validation (Cawley and Talbot, 2010; Parvandeh et al., 2020; Tenenbaum et al., 2000; Varma and Simon, 2006; Zou and Hastie, 2005). We split the dataset into training (9/10) and test (1/10) partitions, and each training partition was further split into inner training and testing folds using another ten-fold cross-validation. Within the inner fold, elastic net regularization finds a set of non-redundant features to explain the dependent variable. Using a linear regression, we predicted the IQ scores of inner fold test data using the features of the selected brain regions by controlling for age, sex, site, and head motion. The model with minimum mean absolute error (MAE) across the inner folds was applied to the test partition of the outer fold, and the IQ scores of outer fold test data were predicted. The prediction procedure was repeated 100 times with different training and test sets to reduce subject selection bias.”

– Interpretability and Lack of ComparisonThe authors claim repeatedly that they are "capitalizing on advanced manifold learning techniques".One could imagine an infinite number of papers that take a dataset, use a technique to extract a metric, X (e.g., eccentricity), and then write about the changes in X with some property of interest, Y (e.g., age). Given this set of papers (and the non-independence between the set of possible Xs), the reader ought to be most interested in those Xs that provide the best performance and simplest interpretation, with other papers being redundant.Thus, a nuanced approach to presenting a paper like this is to demonstrate that the metric used represents an advance over alternative, simpler-to-compute, or clearer-to-interpret metrics that already exist.In this paper, however, the authors do not demonstrate the benefits of their particular choice of applying a specific nonlinear dimensionality reduction method using 3 dimensions alignment to a template manifold and then computing an eccentricity metric.

We thank the reviewers for this suggestion, please see the points below. In brief, both linear and non-linear dimensionality reduction techniques compress high dimensional connectome data (*n x n*) into a series of lower-dimensional eigenvectors (*i.e.,* gradients, each of the form *n x 1*), offering a synoptic view on connectome development. In our case, we used these manifold learning techniques to derive a three-dimensional connectivity space to which we were able to sensitively track developmental changes in adolescence. Beyond these methodological considerations, prior work has shown that principal gradients estimated from task-free functional (Margulies et al., 2016), microstructural (Paquola et al., 2019b), and diffusion MRI (Park et al., 2020a) broadly converge along an established model of sensory-fugal neural hierarchy and laminar differentiation (Mesulam, 1998), allowing the gradient mapping approach to align with theories of cortical organization and evolution (Buckner and Krienen, 2013; Goulas et al., 2018; Huntenburg et al., 2018; Sanides, 1969, 1962). We believe that the relative analytical simplicity and ability to contextualize work in classic theory represent an attractive justification for gradient mapping techniques. Below, we furthermore addressed the specific comments:

For example:i. Is the nonlinearity required (e.g., does it outperform PCA or MDS)?

Non-linear dimensionality techniques are theoretically appealing, as they can explain more information of the original data with the same number of components than linear techniques, which may boost performance on group classification, cluster identification, and phenotypic score prediction (Errity and McKenna, 2007; Gallos et al., 2020; Hong et al., 2020). Of note, diffusion map embedding is analytically straightforward, as it simply performs a singular value decomposition on the Markov matrix, which can be readily interpreted as a diffusion process throughout the connectome. Researchers have previously adopted both linear (Hong et al., 2020; Murphy et al., 2019; Shine et al., 2019; Tian et al., 2020) and non-linear techniques (Guell et al., 2018; Haak and Beckmann, 2020; Hong et al., 2019; Huntenburg et al., 2017; Larivière et al., 2020; Margulies et al., 2016; Müller et al., 2020; Paquola et al., 2019b, 2019a; Sofie L. Valk et al., 2020), and it thus remains an open question as to which technique should be preferred (Hong et al., 2020; Vos de Wael et al., 2020). We opted to stick with the diffusion maps in our main analysis to stay consistent with prior work from our group and others (Hong et al., 2019, 2020; Huntenburg et al., 2017; Larivière et al., 2020; Margulies et al., 2016; Müller et al., 2020; Paquola et al., 2019a; Park et al., 2021; Sofie L. Valk et al., 2020; Vos de Wael et al., 2020). In the revised sensitivity analyses, we also derived gradients via PCA, and showed consistent manifold dimensions using this technique (linear correlation = 0.998 ± 0.001 across E1/E2/E3; Figure 1—figure supplement 7A), as well as consistent age-effects on manifold eccentricity (Figure 1—figure supplement 7B).

We updated Results:

“e) Connectome manifold generation using principal component analysis. In a separate analysis, we generated eigenvectors using principal component analysis (Wold et al., 1987), instead of diffusion map embedding (Coifman and Lafon, 2006), and found consistent spatial maps (linear correlation = 0.998 ± 0.001 across E1/E2/E3; Figure 1—figure supplement 7A) and longitudinal findings (Figure 1—figure supplement 7B).”

Materials and methods:

“An affinity matrix was constructed with a normalized angle kernel, and eigenvectors were estimated via diffusion map embedding (Figure 1A), a non-linear dimensionality reduction technique (Coifman and Lafon, 2006) that projects connectome features into low dimensional manifolds (Margulies et al., 2016). This technique is only controlled by few parameters, computationally efficient, and relatively robust to noise compared to other non-linear techniques (Errity and McKenna, 2007; Gallos et al., 2020; Hong et al., 2020; Tenenbaum et al., 2000), and has been extensively used in the previous gradient mapping literature (Hong et al., 2019, 2020; Huntenburg et al., 2017; Larivière et al., 2020a; Margulies et al., 2016; Müller et al., 2020; Paquola et al., 2019a; Park et al., 2021; Sofie L. Valk et al., 2020; Vos de Wael et al., 2020).”

“e) Connectome manifold generation using principal component analysis. To explore consistency of our results when using different dimensionality reduction techniques, we generated connectome manifolds using principal component analysis (Wold et al., 1987), instead of relying on diffusion map embedding (Coifman and Lafon, 2006), and performed longitudinal modeling (Figure 1—figure supplement 7). We compared the eigenvectors estimated from diffusion map embedding and principal component analysis using linear correlations.”

as well as Discussion:

“In our longitudinal study, we could identify marked connectome expansion during adolescence, mainly encompassing transmodal and heteromodal association cortex in prefrontal, temporal, and posterior regions, the territories known to mature later in development (Gogtay et al., 2004; Shaw et al., 2006). Findings remained consistent when we considered a linear dimensionality reduction technique, suggesting robustness to methodological details of this analysis.”

ii. Is there something special about picking 3 dimensions to do the eccentricity calculation? Is dimensionality reduction required at all (e.g., would you get similar results by computing eccentricity in the full-dimensional space?)

Based on the screen plot in Figure 1A, we selected the first three eigenvectors as (i) they explained approximately 50% of variance in the affinity matrix, (ii) each explained >10% (*i.e.,* 20.7/15.8/13.5% for E1/E2/E3, respectively), and (iii) the choice of three eigenvectors captured a clearly visible eigengap. Prior work has chosen a similar approach, as higher order components may sometimes not show clear spatial patterns and/or higher noise contributions (Haak et al., 2018; Hong et al., 2019, 2020; Margulies et al., 2016; Paquola et al., 2019b, 2019a; Park et al., 2021, 2020a; Vos de Wael et al., 2018). In response to the reviewers’ suggestion, we also assessed manifold eccentricity from the full dimensional space (Figure 1—figure supplement 9A-B). The spatial pattern of manifold eccentricity correlated with the original solution based on three eigenvectors (r = 0.54, p < 0.001). Moreover, longitudinal changes in manifold eccentricity were similar to the original findings (linear correlation of t-statistic map = 0.68, p < 0.001; Figure 1—figure supplement 9C), confirming manifold expansion in transmodal cortices.

We updated Results:

“g) Manifold eccentricity based on all eigenvectors. Repeating manifold eccentricity calculation and age modeling using all eigenvectors, instead of using only the first three, we observed relatively consistent results with our original findings (linear correlation of manifold eccentricity r = 0.54, p < 0.001; t-statistic map r = 0.68, p < 0.001), also pointing to manifold expansion in transmodal cortices (Figure 1—figure supplement 9).”

as well as Materials and methods:

“g) Manifold eccentricity analysis based on all eigenvectors. We repeated our analysis by calculating manifold eccentricity from all eigenvectors to assess consistency of the findings (Figure 1—figure supplement 9).”

iii. Does it outperform basic connectome measures (e.g., the simple ones the authors compute)?

To compare the age-effects on manifold eccentricity with graph-theoretical effects, we calculated betweenness, degree, and eigenvector centrality measures (Rubinov and Sporns, 2010). While betweenness centrality did not reveal significant effects, degree and eigenvector centrality showed effects in similar regions as manifold eccentricity (Figure 1—figure supplement 8). Calculating linear correlation between the effect size maps, we found a graded pattern of similarity, which was low but significant for betweenness centrality (r = 0.18, p = 0.02; significance determined by 1,000 spin tests that account for spatial autocorrelation (Alexander-Bloch et al., 2018)) and moderate for degree and eigenvector centrality (r = 0.57, p < 0.001; r = 0.47, p < 0.001).

We updated Results:

“f) Longitudinal changes in graph-theoretical measures. Repeating the longitudinal modeling using graph-theoretical centrality measures, we found significant age-related longitudinal changes in degree and eigenvector centrality, while betweenness centrality did not reveal significant effects, in similar regions to those that had significant age-related changes in manifold eccentricity (Figure 1—figure supplement 8). Correlating the effect size maps for manifold eccentricity and each graph measure, we found a significant yet variable spatial similarity of the effect maps (betweenness centrality: r = 0.18, spin-test p = 0.02; degree centrality: r = 0.57, p < 0.001; eigenvector centrality: r = 0.47, p < 0.001).”

as well as Materials and methods:

“f) Longitudinal changes in graph-theoretical measures. To compare longitudinal changes in manifold eccentricity with those in graph-theoretical centrality measures, we calculated betweenness, degree, and eigenvector centrality of the structural connectomes and build similar linear mixed effects models to assess longitudinal change (Figure 1—figure supplement 8). Betweenness centrality is the number of weighted shortest paths between any combinations of nodes that run through that node, degree centrality is the sum of edge weights connected to a given node, and eigenvector centrality measures the influence of a node in the whole network (Lohmann et al., 2010; Rubinov and Sporns, 2010; Zuo et al., 2012). Spatial similarity between t-statistics of centrality and manifold measures were assessed with 1,000 spin tests that account for spatial autocorrelation (Alexander-Bloch et al., 2018).”

There is a clear down-side of how opaque the approach is (and thus difficult to interpret relative to, say, connectivity degree), so one would hope for a correspondingly strong boost in performance. The authors could also do more to develop some intuition for the idea of a low-dimensional connection-pattern-similarity-space, and how to interpret taking Euclidean distances within such a space.CT=1N[∑i=1NT(E1)i,∑i=1NT(E2)i,∑i=1NT(E3)i]ME=∑e=13{I(Ee)−CT(e)}2

Here, we used eigenvectors estimated from the decomposition technique to generate a new coordinate system in connectivity space (Bijsterbosch et al., 2020; Haak et al., 2018; Huntenburg et al., 2018; Margulies et al., 2016; Mars et al., 2018). Notably, in this manifold space, interconnected brain regions with similar connectivity patterns are closer together, while regions that do not have significant connectivity nor similarity in connectivity patterns are located farther apart. In our work, the specific space is a diffusion map, where Euclidean distances in the manifold correspond to diffusion times between the nodes of the network. To analyze how the multi-dimensional manifold structures change in the low dimensional manifold space, we simplified the multivariate eigenvectors into a single scalar value *i.e.,* manifold eccentricity. Manifold eccentricity was calculated as the Euclidean distance between the manifold origin and all data points (*i.e.,* brain regions) in manifold space. For example, using three eigenvectors, manifold eccentricity was defined as follows:CT is the template manifold origin; N number of brain regions; T(•) template manifold; ME manifold eccentricity; I(•) individual manifold; CT(e) origin of e^th^ template manifold. This concept is visualized in Figure 1—figure supplement 15. Each brain region (*i.e.,* each dot in the scatter plot) is described as a vector from the manifold origin (*i.e.,* triangular mark in the scatter plot), and manifold eccentricity is simply a length (*i.e.,* Euclidean distance) of that vector. Shifts in connectivity patterns of a given region thus will lead to shifts in the vectors, which in turn changes the manifold eccentricity. Thus, manifold eccentricity quantifies global brain organization in connectivity space.

We updated Materials and methods:

“Briefly, the eigenvectors estimated from the decomposition technique generate a connectivity coordinate system (Bijsterbosch et al., 2020; Haak et al., 2018; Huntenburg et al., 2018; Margulies et al., 2016; Mars et al., 2018) – the diffusion map, where […] Shifts in connectivity patterns of a given region thus will lead to shifts in the vectors, which in turn changes the manifold eccentricity. Thus, manifold eccentricity quantifies global brain organization based in the connectivity space.”

– Developmental Enrichment AnalysisBoth in the main text and in the Materials and methods, this is described as "genes were fed into a developmental enrichment analysis". Can some explanation be provided as to what happens between the "feeding in" and what comes out? Without clearly described methods, it is impossible to interpret or critique this component of the paper. If the methodological details are opaque, then the significance of the results could be tested numerically relative to some randomized null inputs being 'fed in' to demonstrate specificity of the tested phenotype.

We implemented developmental enrichment analysis using the cell-type specific expression analysis (CSEA) developmental expression tool (http://genetics.wustl.edu/jdlab/csea-tool-2) (Dougherty et al., 2010; Xu et al., 2014). This technique evaluates the significance of the overlap between the identified gene list and RNAseq data obtained from BrainSpan dataset (http://www.brainspan.org) across six brain regions (*i.e.,* cortex, thalamus, striatum, cerebellum, hippocampus, amygdala) and ten developmental periods (from early fetal to young adulthood) approximated from mouse data, yielding a total of 60 combinations of developmental enrichment profiles (Xu et al., 2014). The significance is calculated based on Fisher’s exact tests (Fisher, 1922) with FDR correction (Benjamini and Hochberg, 1995).

We updated Results:

“We performed developmental gene set enrichment analysis using the cell-type specific expression analysis (CSEA) tool, which compares the selected gene list with developmental enrichment profiles (see Methods) (Dougherty et al., 2010; Xu et al., 2014). This analysis highlights developmental time windows across macroscopic brain regions in which genes are strongly expressed. We found marked expression of the genes enriched from childhood onwards in the cortex, thalamus, and cerebellum (FDR < 0.001; Figure 4B).”

as well as Materials and methods:

“Leveraging cell-type specific expression analysis (CSEA) developmental expression tool (http://genetics.wustl.edu/jdlab/csea-tool-2) (Dougherty et al., 2010; Xu et al., 2014), we evaluated the significance of overlap between the genes showing consistent whole-brain expression pattern across donors (FDR < 0.05) with RNAseq data obtained from BrainSpan dataset (http://www.brainspan.org). The significance was calculated based on Fisher’s exact test (Fisher, 1922) with FDR correction (Benjamini and Hochberg, 1995). The CSEA tool provides simplified results of gene enrichment profiles along six major brain regions (i.e., cortex, thalamus, striatum, cerebellum, hippocampus, amygdala) across ten developmental periods (from early fetal to young adulthood) approximated from mouse data, yielding a total of 60 combinations of developmental enrichment profiles (Xu et al., 2014).”

As suggested, we additionally assessed spatial correlations between the *post-mortem* gene expression maps and rotated maps of the age-related changes in manifold eccentricity (100 spherical, random rotations). For each iteration, the selected genes were used for developmental enrichment analysis using CSEA tool (Xu et al., 2014), and we obtained FDR-corrected p-values for each brain division and developmental period. We, thus, built a null distribution using the rotated p-values into which the actual p-value was placed. If the actual p-value did not belong to the 95% of the null distribution, it was deemed significant. The results remained consistent.

We updated Materials and methods:

“We repeated developmental enrichment analysis using the genes identified from the rotated maps of the age-related changes in manifold eccentricity (100 spherical rotations). For each iteration, we obtained developmental expression profiles using the identified genes, where the FDR-corrected p-values built a null distribution. For each brain division and developmental period, if the actual p-value placed outside 95% of the null distribution, it was deemed significant.”

– IQ predictionThe predictions seem to be very poor (equality lines, y = x, should be drawn in Figure 5, to show what perfect predictions would look like; linear regressions are not helpful for a prediction task, and are deceptive of the appropriate MAE computation). The authors do not perform any comparisons in this section (even to a real baseline model like `predicted_IQ = mean(training_set_IQ)`). They also do not perform statistical tests (or quote p-values), but nevertheless make a range of claims, including of "significant prediction" or "prediction accuracy was improved", "reemphasize the benefits of incorporating subcortical nodes", etc. All of these claims should be tested relative to rigorous statistics, and comparisons to appropriate baseline/benchmark approaches.

We thank the reviewer for the comment. To assess the significance of our prediction model, we compared the prediction performance to the suggested baseline model (*i.e.,* predicted IQ = mean(training set IQ)). The prediction performance for the baseline model was not good showing a low negative correlation between actual and predicted IQ scores (r = -0.15 ± 0.06, MAE = 8.98 ± 0.04, p = 0.12). Our model based on cortical features (baseline: r = 0.14 ± 0.04, MAE = 8.93 ± 0.16, p = 0.09; baseline and maturational change: r = 0.18 ± 0.04, MAE = 9.10 ± 0.19, p = 0.04), and based on both cortical and subcortical features (baseline: r = 0.17 ± 0.03, MAE = 8.74 ± 0.11, p = 0.04; baseline and maturational change: r = 0.21 ± 0.02, MAE = 8.86 ± 0.14, p = 0.01) outperformed the baseline model (Meng’s z-test p < 0.001 for all cases) (Meng et al., 1992). The results suggest that while manifold features are only weakly related to IQ, they still provide information on future IQ above and beyond the baseline model.

We updated the Results:

“We compared the prediction performance of our model with a baseline model, where IQ of the test set was simple average of training set (r = -0.15 ± 0.06, MAE = 8.98 ± 0.04, p = 0.12; see Methods). We found that our model outperformed this baseline model (Meng’s z-test p < 0.001) (Meng et al., 1992).”

Materials and methods:

“To assess whether our model outperforms baseline model, we predicted IQ of test data using average of IQ of training data (i.e., predicted IQ = mean(training set IQ)). The improvement of prediction performance was assessed using Meng’s z-test (Meng et al., 1992).”

as well as Discussion:

“Of note, although our model significantly outperformed a baseline model, the relationship between the actual and predicted IQ scores did not locate on the equality line and the strength of the association was rather weak. Further improvements in brain-based IQ prediction in adolescence, for example through combinations of structural and functional imaging features, will be a focus of future work.”

We additionally performed the prediction analysis using a regression tree method (*i.e.,* decision tree learning), a non-linear approach based on tree model constituting root node and split leaf nodes, where the leaf nodes contain the response variables (Breiman et al., 1984). However, the prediction results were not improved compared to linear model (Figure 5—figure supplement 2).

We updated Results:

“l) IQ prediction using non-linear model. We predicted IQ at follow-up using a regression tree method (Breiman et al., 1984), instead of linear regression model, but we could not find improved prediction performance (Figure 5—figure supplement 2).”

as well as Materials and methods:

“l) IQ prediction using non-linear model. We additionally performed prediction analysis for predicting future IQ score using regression tree method (i.e., decision tree learning), a non-linear approach based on tree model constituting root node and split leaf nodes, where the leaf nodes contain the response variables (Breiman et al., 1984) (Figure 5—figure supplement 2).”

– Group ConnectomeGiven how much the paper relies on estimating a group structural connectome, it should be visualized and characterized. For example, a basic analysis of the distribution of edge weights and degree, especially as edge weights can vary over orders of magnitude and high weights (more likely to be short distances) may therefore unduly dominate some of the low-dimensional components). The authors may also consider testing robustness performed to alternative ways of estimating the connectome [e.g., Oldham et al. NeuroImage 222, 117252 (2020)] and its group-level summary [e.g., Roberts et al. NeuroImage 145, 1-42 (2016)].

We thank the Reviewers for these suggestions. We calculated structural connectome in individual level, and estimated group representative connectome using a distance-dependent thresholding (Betzel et al., 2019), which is visualized in Figure 1A. As suggested, we performed the longitudinal modeling using the connectome edge weights and found significant increases in weights within frontoparietal and default mode networks, as well as attention and sensory networks (FDR < 0.05; Figure 1—figure supplement 12). These results are consistent with our main findings based on manifold eccentricity that pointed to mainly transmodal effects.

We updated Results:

“j) Longitudinal modeling using edge weights. Repeating the longitudinal modeling across age using connectome edge weights, we found significant increases in edge weights within frontoparietal and default mode networks, as well as in attention and sensory networks (FDR < 0.05; Figure 1—figure supplement 12), consistent with findings based on manifold eccentricity.”

as well as Materials and methods:

“j) Longitudinal modeling using edge weights. In addition to the analyses based on manifold eccentricity, linear mixed effect modeling using connectome edge weights assessed age-related longitudinal changes in streamline strength (Figure 1—figure supplement 12).”

To assess robustness of group representative structural connectomes, we built the connectomes with two different approaches: (i) Distance-dependent thresholding (Betzel et al., 2019) was adopted for the main analysis, and (ii) consistency thresholding was assessed in a supplementary analysis. Consistency thresholding approach averages subject specific matrices, in addition to performing a 50, 40, 30, 20, 10% thresholding, as well as simple averaging (*i.e.,* 0% thresholding) (Wang et al., 2019) (Figure 1—figure supplement 10). We calculated linear correlations between spatial maps of eigenvectors derived from distance-dependent thresholding and the group consistency method and found high similarity (r = 0.89 ± 0.01 for E1; 0.93 ± 0.004 for E2; 0.85 ± 0.01 for E3 across consistency thresholds).

We updated Results:

“h) Robustness of group representative structural connectome. We compared gradients derived from the group representative structural connectome, based on (i) distance-dependent thresholding (Betzel et al., 2019) and (ii) consistency thresholding (Wang et al., 2019) (Figure 1—figure supplement 10). We found high similarity in spatial maps of the estimated manifolds (r = 0.89 ± 0.01 for E1; 0.93 ± 0.004 for E2; 0.85 ± 0.01 for E3 across six different thresholds), indicating robustness.”

as well as Materials and methods:

“h) Robustness of group representative structural connectome. We compared the distance-dependent thresholding (Betzel et al., 2019) that was adopted for the main analysis with a consistency thresholding approach (Wang et al., 2019). The latter averages subject specific matrices, in addition to performing a 50, 40, 30, 20, 10% thresholding, as well as simple averaging (i.e., 0% thresholding) (Figure 1—figure supplement 10).”

*–* Individual Alignment

The paper relies on individuals being successfully aligned to the template manifold. Accordingly, some analysis should be performed quantifying how well individuals could be mapped. Presumably some subjects fit very well onto the template, whereas others do not. Is there something interesting about the poorly aligned subjects? Do your results improve when excluding them?

Individual manifolds were aligned to template manifold with Procrustes alignment, which makes eigenvectors from different individuals more comparable *e.g.,* by flipping eigenvector signs (Langs et al., 2015; Vos de Wael et al., 2020). To evaluate the procedure, we calculated linear correlations between template and individual manifolds before/after alignment. We found significant improvement in correlations after aligning individual manifolds to the template (0.92±0.03/0.93±0.03/0.94±0.03 after alignment; -0.02±0.03/-0.001±0.37/0.003±0.12 before alignment for E1/E2/E3).

As suggested, we also performed the longitudinal modeling after excluding 10% of subjects with poor alignment (cutoff r = 0.83; the new set had a linear correlation with template manifold of mean ± SD r = 0.94 ± 0.01). We still could find consistent results (Figure 1—figure supplement 6), where the t-statistics showed strong correlation with those derived using the whole subjects (r = 0.97, p < 0.001), suggesting robustness.

We updated Results:

“d) Gradient alignment fidelity. When calculating linear correlations between template and individual manifolds before and after alignment, we found significant increases after alignment (r = 0.92±0.03/0.93±0.03/0.94±0.03) compared to before alignment (-0.02±0.03/-0.001±0.37/0.003±0.12), for E1/E2/E3, respectively, supporting effectiveness of alignment. After excluding 10% of subjects with poor alignment (cutoff r = 0.83; the new set was correlated with the template manifold, r = 0.94 ± 0.01), we found consistent age-related changes in manifold eccentricity (Figure 1—figure supplement 6), with the t-statistic map showing strong correlation to the map derived in the whole sample (r = 0.97, p < 0.001).”

as well as Materials and methods:

“d) Gradient alignment fidelity. To assess robustness of individual alignment, we computed linear correlations between the template and individual manifolds before and after alignment. We also repeated the linear mixed effect modeling after excluding 10% of subjects with the lowest alignment to the template manifold (Figure 1—figure supplement 6).”